# The Perception–Physics Paradox: Probing Scientific Alignment with TC-BENCH

**Dingling Yao** [1]   **Andrea Polesello** [1]   **Adeel Pervez** [1]   **Caroline Muller** [1]   **Francesco Locatello** [1]

## Abstract

While Vision Foundation Models (VFMs) excel at predictive tasks on satellite imagery, their performance can arise from visual correlations rather than underlying structural invariants, making even perception-based out-of-distribution accuracy a poor proxy for scientific utility. As a result, models may look correct without reasoning correctly—a discrepancy we term the Perception–Physics Paradox. To address this gap, we introduce scientific alignment as an implicit objective for representation learning in scientific domains. We study a principled, testable aspect of scientific alignment through structural isomorphism, which requires latent representations to uniquely identify physical systems up to a linear reparameterization. This perspective induces a hierarchy of necessary conditions and yields a systematic probing protocol for physical and causal interpretability. To operationalize this framework, we release TC-BENCH, a global, reproducible benchmark dataset with an automated construction pipeline for tropical cyclone research, and show that current VFMs rely on visual shortcuts that collapse in intense regimes, indicating that scientific alignment does not arise as a natural byproduct of scaling alone. We provide all evaluation code and the reproducible data construction pipeline at https://github.com/CausalLearningAI/tc-bench.

## 1. Introduction

Recent vision foundation models exhibit strong zero-shot performance across a wide range of visual benchmarks, spanning recognition, retrieval, and open-ended image and video understanding (Wiedemer et al., 2025; Karypidis et al., 2025; Siméoni et al., 2025; Tschannen et al., 2025; Meng et al., 2024; Radford et al., 2021; Oquab et al., 2023). These successes are often interpreted as evidence that such models acquire broadly transferable representations, motivating their application to scientific domains (Moor et al., 2023; Hasson et al., 2025; Mai et al., 2024).

While prior work has established a strong foundation for applying vision models to scientific domains, evaluations have largely emphasized broad benchmarks and average-case performance (Hasson et al., 2025; Kitamoto et al., 2023). For example, the Digital Typhoon dataset (Kitamoto et al., 2023) provided a vital first benchmark for vision-based meteorology. However, tropical cyclone samples naturally concentrate around moderate, stable regimes (e.g., minimum central pressure near 1000 hPa). As a result, strong average performance can reflect reliance on stable visual regularities, while masking failures in physically extreme regimes, which are often insufficiently resolved by aggregate evaluations (Hasson et al., 2025).

To mitigate this limitation, the community has increasingly turned to out-of-distribution (OOD) generalization as a proxy for assessing whether models learn invariant, physically meaningful representations. However, not all OOD shifts are equally diagnostic. In particular, shifts that primarily alter visual context—such as geographic region or agency-specific conventions—may leave the underlying physical axes insufficiently perturbed. As a result, such tests can give an overly optimistic picture of physical invariance, even when representations remain dominated by visual cues rather than meaningful physical coordinates (see Fig. 1).

> **The Perception–Physics Paradox**
>
> A representation can generalize perceptually and predict physical targets, yet collapse the physical degrees of freedom required for scientific reasoning.

To systematically assess this tension, we introduce *scientific alignment* as a high-level goal for representation learning in scientific tasks. Because this concept is inherently broad, we focus on a principled and testable aspect: *structural isomorphism* (Defn. 2.3). A representation is structurally isomorphic to a physical system if it identifies the system up to a linear reparameterization, allowing physical state variables to be encoded in a distributed—though not nec-

[1]Institute of Science and Technology Austria. Correspondence to: Dingling Yao <dingling.yao@ista.ac.at>.

*Proceedings of the 43rd International Conference on Machine Learning*, Seoul, South Korea. PMLR 306, 2026. Copyright 2026 by the author(s).

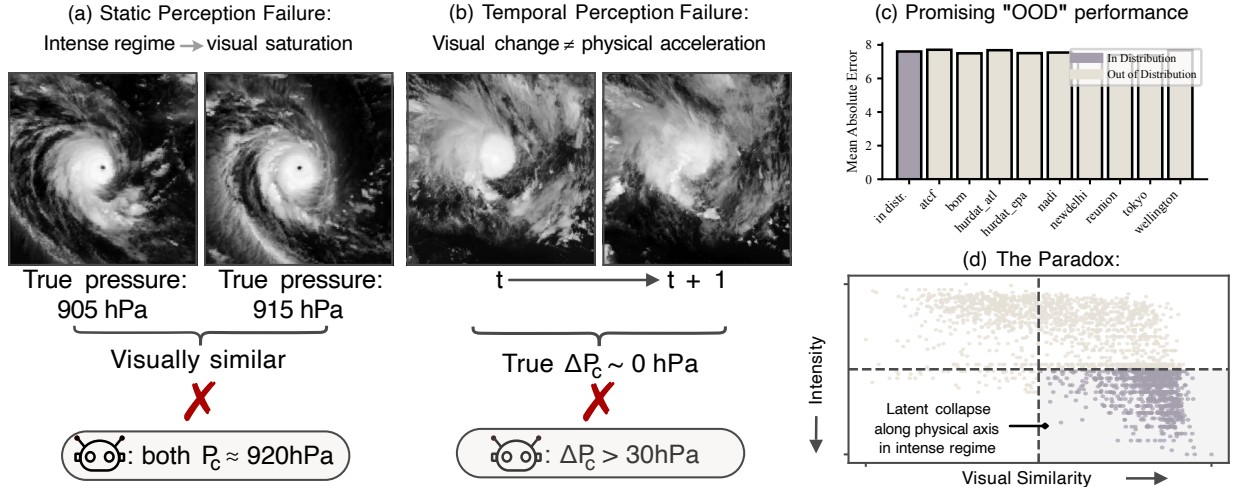

Figure 1. **The Perception-Physics Paradox**: (a) In intense regimes ($P_c < 980$ hPa), visually near-identical cyclones (e.g., 905 vs. 915 hPa) exhibit indistinguishable eye and cloud structures, causing vision models to map them to similar latent states and fail to resolve physically meaningful pressure differences. (b) Visually salient temporal changes (e.g., cloud expansion) can induce large predicted intensity shifts even when the true physical evolution is negligible (true $\Delta P_c = 0$ hPa vs. predicted $\Delta P_c > 30$ hPa).(c) Cross-agency evaluation—training on all but one meteorological agency and testing on the held-out agency—shows strong out-of-distribution performance, suggesting robust visual generalization. (d) Despite this perceptual robustness, latent representations collapse along critical physical axes in intense regimes, demonstrating that perceptual generalization alone does not guarantee physically meaningful representation structure.

essarily disentangled—manner that remains coherent across regimes and system evolution. This formulation induces a hierarchy of necessary conditions, yielding a systematic probing protocol that operationalizes scientific alignment as a set of falsifiable hypotheses about the physical and causal structure encoded in learned representations.

To probe these necessary conditions in a realistic and high-stakes setting, we consider *tropical cyclones*[1] as a diagnostic case. Tropical cyclones are governed by well-defined physical state variables, such as minimum central pressure and maximum sustained wind, which directly determine their destructive potential (Bell et al., 2000). At the same time, these variables are not directly observable at scale: infrared satellite imagery is abundant and inexpensive, whereas *In-situ* measurements of atmospheric variables are often sparse, especially over the oceans, while satellites provide full coverage and high-resolution data (Balsamo et al., 2018). This mismatch makes cyclones a natural setting in which to ask whether visual representations encode physically meaningful states, rather than merely appearance-level correlates.

To this end, we introduce TC-Bench, to our knowledge the first *versioned, global* tropical cyclone benchmark dataset with a unified and reproducible data construction pipeline. By addressing long-standing issues of geographic

coverage, maintenance, and reproducibility in existing resources (Knapp & Kossin, 2007; Knapp, 2008a; Kitamoto et al., 2023), TC-Bench provides a foundation for future data-driven studies of cyclone dynamics and prediction. In this work, however, we do not position TC-Bench as a leaderboard-style benchmark for optimizing predictive performance. Instead, we use it as a *diagnostic testbed* to probe representation quality under *physically targeted stress*. In contrast to perception-based OOD tests that primarily vary visual context or acquisition conditions, we explicitly evaluate representations along critical physical axes, most notably cyclone intensity. We ask whether leading vision foundation models encode latent spaces that are structurally isomorphic to the underlying physical state variables, particularly in intense regimes where visual cues saturate but physical variation remains substantial. This use of TC-Bench disentangles perceptual robustness from the structural requirements needed for scientific reasoning, enabling falsifiable tests of where and how learned representations fail to support physically grounded inference. We summarize our contributions as follows:

- **Scientific Alignment as a Diagnostic Lens.** We articulate the *Perception–Physics Paradox*, highlighting a systematic gap between perceptual robustness and the utility of learned representations as physical state variables. To study this gap, we introduce *scientific alignment* (§ 2) as an evaluative lens and formalize a set of *necessary, falsifiable conditions* through the notion of *structural isomorphism*. This perspective characterizes

---

[1]"Tropical cyclones" are intense, rotating storms that develop over the warm waters of the tropical oceans. They are classified by wind speed as depressions, storms, or cyclones ($\geq$118 km/h), with the latter regionally termed "hurricanes" in North Atlantic Ocean or "typhoons" in the Western Pacific (Emanuel, 2003).

when representations are consistent with serving as structural surrogates of physical systems, rather than relying solely on appearance-level correlations.

- **A Global Benchmark for Tropical Cyclone Research.** We introduce TC-BENCH, a versioned, **global** tropical cyclone benchmark dataset with a standardized and reproducible data construction pipeline (§ 3). By integrating multi-agency metadata and satellite observations across all major basins, TC-BENCH supports long-term, reproducible studies of cyclone dynamics. Crucially, it explicitly includes intense regimes and life-cycle transitions, enabling physically targeted representational stress tests that are invisible under conventional average-case evaluations.

- **A Unified Probing Framework for Scientific Representations.** We propose a systematic probing framework derived from the necessary conditions of scientific alignment (§ 2). Applying these probes to leading vision foundation models, we show that representations that perform well under standard in-distribution and OOD benchmarks can nonetheless collapse in physically extreme regimes, where visual similarity masks meaningful physical variation. Together, these probes form a principled diagnostic toolkit for localizing *where* and *how* learned representations fail to support physically grounded scientific reasoning.

**Scope and Positioning.** *Our goal is diagnostic rather than prescriptive.* We do not claim that vision foundation models are inherently incapable of learning physically aligned representations, nor that nonlinear probes or task-specific training cannot recover useful physical information. Rather, we show that commonly used evaluation protocols—based on average-case performance or perception-driven OOD shifts—are insufficient to certify alignment with respect to specific scientific queries. To make this failure mode measurable, we introduce query-aware, physically grounded probes that provide falsifiable tests of representational structure. Under these tests, frozen VFM representations fail to satisfy even weak necessary conditions of structural alignment in physically extreme regimes. TC-BENCH therefore serves as a controlled stress test for a broader class of settings in which observations remain visually coherent while becoming locally insensitive to physically important variation.

## 2. The Scientific Alignment Framework

Deep learning has proven remarkably effective at learning compressed representations of complex data; however, in scientific domains, compression alone is often insufficient (Roscher et al., 2020). For a model to be useful for discovery, its internal representations must satisfy at least minimal requirements of **scientific alignment**, which we interpret here through one concrete, testable aspect: whether the learned latent space is *structurally isomorphic*

to the physical state space (Defn. 2.3). While a universal definition of *scientific alignment* may be challenging to provide, we propose a set of necessary conditions that focus on structural and functional correspondence between a learned encoding and its target physical system (Swoyer, 1991). Under this view, an aligned representation acts as a *structural surrogate* (Cummins, 1989), preserving physically meaningful variation across regimes. We proceed by formalizing structural isomorphism, deriving its uniform implications, and introducing empirical probes that estimate the corresponding residuals, culminating in an interventional consistency guarantee (Thm. 2.1).

**Definition 2.1** (Physical System). A physical system $\mathcal{S}$ represents a governing process $\mathbf{y} \in \mathcal{Y} \subset \mathbb{R}^m$ observed as data $\mathbf{x} \in \mathcal{X}$ across diverse regimes $\mu \in \mathcal{M}$. It is defined by:

(i) *Governing Dynamics*: A vector field $\mathbf{f}_\mu$ governing system evolution, written as $\dot{\mathbf{y}} = \mathbf{f}_\mu(\mathbf{y})$, with discrete-time observations treated as finite differences.

(ii) *Observation Process*: A mapping $\phi$ where $\mathbf{x} = \phi(\mathbf{y})$, defining how we measure the state.

(iii) *Invariants*: A set of constraints $\mathcal{P}(\mathbf{y}) = 0$ (e.g., conservation laws) that define the valid physical manifold.

Each regime $\mu \in \mathcal{M}$ induces a data distribution $\mathcal{D}_\mu$ over trajectories and observations through the governing dynamics $\mathbf{f}_\mu$ and observation process $\phi$. All properties of $\mathcal{S}$ are required to hold uniformly across the regime family $\mathcal{M}$.

**Definition 2.2** (Representation). Let $g : \mathcal{X} \to \mathcal{Z} \subset \mathbb{R}^k$ be a learned encoder mapping observations to a latent space; we define $\mathbf{z} = g(\mathbf{x})$ as the *learned representation* of the observation $\mathbf{x} \in \mathcal{X}$.

### 2.1. Formalism: Structural Isomorphism

While Scientific Alignment represents a broad epistemic goal, i.e., ensuring that a model's internal logic corresponds to physical reality—it requires a formal interpretation to be empirically testable. Accordingly, we examine one concrete aspect of scientific alignment: whether a learned representation exhibits *structural isomorphism* with the physical system, which we operationalize as follows:

**Definition 2.3** (Structural Isomorphism). A representation $\mathbf{z} = g(\mathbf{x}) \in \mathcal{Z}$ is *structurally isomorphic* to a physical system $\mathcal{S}$ if there exists an injective linear map $\mathbf{A} \in \mathbb{R}^{d \times m}$ such that, for all regimes $\mu \in \mathcal{M}$,

$$\mathbf{z} = \mathbf{A}\mathbf{y} + \epsilon_\mu(\mathbf{y}), \tag{2.1}$$

where the residual $\epsilon_\mu(\mathbf{y})$ is uniformly bounded in magnitude and variation:

$$\sup_\mu \mathbb{E}_{\mathbf{y} \sim \mathcal{D}_\mu}[\|\epsilon_\mu(\mathbf{y})\|] \leq \bar{\epsilon},$$
$$\sup_\mu \mathbb{E}_{\mathbf{y} \sim \mathcal{D}_\mu}[\|J_{\epsilon_\mu}(\mathbf{y})\|] \leq \bar{\delta}, \tag{2.2}$$

where $J_{\epsilon_\mu}(\mathbf{y}) := \frac{\partial \epsilon_\mu}{\partial \mathbf{y}}$ and $\|\cdot\|$ denotes the operator norm.

*Remark* 2.1 (***Distributed Structural Identifiability***). Under this criterion, a representation $g$ is aligned if its latent space identifies the physical system $\mathcal{S}$ as a unique, linearly reparameterized subspace. Unlike traditional CRL, which often requires coordinate-wise disentanglement (Schölkopf et al., 2021; von Kügelgen et al., 2024), structural isomorphism allows for a *distributed representation*. Even if there is no element-wise correspondence between $\mathbf{z}$ and the physical state variables $\mathbf{y}$, Defn. 2.3 ensures that $g$ identifies a unique physical system $\mathcal{S}$ within a linear subspace. This form of structural preservation reflects sensitivity to the geometry of the governing dynamics $\mathbf{f}_\mu$ rather than reliance on superficial statistical correlations.

*Remark* 2.2 (***Interpretation and Scope***). The linear reparameterization in Defn. 2.3 does not assume linear physics, nor does it imply that deep representations are globally linear. Rather, it tests whether the latent space preserves a coordinate system that is *locally consistent across the regimes under study*. Structural isomorphism should therefore be understood as an approximate and falsifiable hypothesis about latent geometry: failure indicates misalignment, while success constitutes a necessary—but not sufficient—condition for scientific alignment.

**Proposition 2.1** (Uniform Implications of Structural Isomorphism). *Given a physical system $\mathcal{S}$ with bounded vector field $\|\mathbf{f}_\mu(\mathbf{y})\| \leq K$ for all regime $\mu \in \mathcal{M}$, if a representation $\mathbf{z} = g(\mathbf{x})$ is structurally isomorphic to $\mathcal{S}$ (Defn. 2.3) and the constraints $\mathcal{P}$ in $\mathcal{S}$ are $\Lambda_\mathcal{P}$-Lipschitz continuous, then there exists a linear decoder $L \in \mathbb{R}^{m \times d}$ (e.g. a left inverse of $\mathbf{A}$) such that, uniformly over the regime family $\mathcal{M}$, the following hierarchical properties hold:*

*(i) Static Fidelity. The physical state is uniformly recoverable from the representation:*

$$\sup_{\mu \in \mathcal{M}} \mathbb{E}_{\mathbf{y} \sim \mathcal{D}_\mu} \left[ \|\mathbf{y} - L\mathbf{z}\| \right] \leq \|L\|\bar{\epsilon} =: \epsilon_{\text{stat}}. \quad (2.3)$$

*(ii) Dynamic Coherence. The decoded latent derivatives $\dot{\mathbf{z}}$ uniformly approximate the physical derivatives $\dot{\mathbf{y}}$:*

$$\sup_{\mu \in \mathcal{M}} \mathbb{E}_{(\mathbf{y}, \dot{\mathbf{y}}) \sim \mathcal{D}_\mu} \|\dot{\mathbf{y}} - L\dot{\mathbf{z}}\| \leq \|L\|\bar{\delta}K =: \epsilon_{\text{dyn}},$$

$$(2.4)$$

*where $L$ is the same decoder as in (i).*

*(iii) Manifold Consistency. The linear reconstruction approximately satisfies the system invariants $\mathcal{P}$:*

$$\sup_{\mu \in \mathcal{M}} \mathbb{E}_{\mathbf{y} \sim \mathcal{D}_\mu} \left[ \|\mathcal{P}(L\mathbf{z})\| \right] \leq \Lambda_\mathcal{P}\|L\|\bar{\epsilon} =: \epsilon_{\text{con}}. \quad (2.5)$$

### 2.2. Operationalization: Structural Alignment Probes

Prop. 2.1 characterizes the necessary, error-bounded consequences of structural isomorphism at the population level.

*Table 1.* **Structural Alignment Hierarchy.** Each probe $\mathcal{Q} = (\mathcal{Z}, \mathcal{R}, \mathcal{H}, \psi)$ measures a uniform residual $\epsilon$ across regimes $\mathcal{M}$, serving as a necessary condition for scientific alignment. In practice, we approximate the derivatives in $\mathcal{Q}_{\text{dyn}}$ with finite difference over fixed time span: $\Delta\mathbf{z} := L(\mathbf{z}_{t+1} - \mathbf{z}_t), \Delta\mathbf{y} = \mathbf{y}_{t+1} - \mathbf{y}_t$.

| Probe | Ref. $\mathcal{R}$ | Surr. $\mathcal{H}$ | Metric $\psi(\cdot)$ |
|---|---|---|---|
| $\mathcal{Q}_{\text{STAT}}$ | $\mathbf{y}$ | $L$ | $\|\mathbf{y} - L\mathbf{z}\|$ |
| $\mathcal{Q}_{\text{DYN}}$ | $d\mathbf{y}$ | $L$ | $\|\Delta\mathbf{y} - L\Delta\mathbf{z}\|$ |
| $\mathcal{Q}_{\text{CON}}$ | $\mathcal{P}$ | $L$ | $\|\mathcal{P}(L\mathbf{z})\|$ |

In practice, however, structural isomorphism is only approximate and cannot be verified directly. This motivates a family of empirical diagnostics that estimate the tightest achievable residuals under restricted surrogate models. To this end, we introduce the *Structural Alignment Probe*:

**Definition 2.4** (Structural Alignment Probe). A *Structural Alignment Probe* is a formal diagnostic quadruplet $\mathcal{Q} = (\mathcal{Z}, \mathcal{R}, \mathcal{H}, \psi)$ that interrogates a learned representation $\mathbf{z} = g(\mathbf{x})$ by seeking a restricted-capacity surrogate $h \in \mathcal{H}$ that minimizes the *worst-case discrepancy* across the regime family $\mathcal{M}$. The probe value $V(g)$ is defined as the uniform residual:

$$V(g) = \inf_{h \in \mathcal{H}} \sup_{\mu \in \mathcal{M}} \mathbb{E}_{(\mathbf{z}, \mathbf{r}) \sim \mathcal{D}_\mu} \left[ \psi(h(\mathbf{z}), \mathbf{r}) \right] \quad (2.6)$$

where $h(\mathbf{z})$ is the predicted variable, $\mathbf{r} \in \mathcal{R}$ is the physical reference, $\psi$ is the error metric. We say that $g$ passes the probe $\mathcal{Q}$ at tolerance $\epsilon$ if $V(g) \leq \epsilon$.

Conceptually, a structural alignment probe quantifies how well a learned representation supports the error bounds implied by structural isomorphism when accessed through a restricted surrogate class $\mathcal{H}$. Instantiating this definition using the implications in Prop. 2.1 yields a hierarchy of probes targeting state recoverability, dynamical coherence, and consistency with known physical constraints. Together, these probes act as *falsifiable tests* of whether a latent representation is consistent with preserving the geometry of the physical state manifold.

### 2.3. From Structural Isomorphism to Causal Reasoning

**Theorem 2.1** (Interventional Grounding). *Let $\mathcal{Y}$ be a compact manifold and assume a bounded physical vector field $\|\mathbf{f}_\mu\| \leq K$ for all $\mu \in \mathcal{M}$. Let $g : \mathcal{X} \to \mathcal{Z}$ be an encoder that satisfies the structural alignment hierarchy (Prop. 2.1) with uniform residuals $\{\epsilon_{stat}, \epsilon_{dyn}\}$ under a linear decoder $L$. Let $G = g \circ \phi$ denote the induced mapping from the state space $\mathcal{Y}$ to the latent space $\mathcal{Z}$. Then, for a hard intervention $do(\mathbf{y}^*)$ applied at time $t^*$, the $n$-step interventional rollout satisfies*

$$\|L\Delta^n G(\mathbf{y}^*) - \Delta^n(\mathbf{y}^*)\| \leq \epsilon_{int}(n), \quad (2.7)$$

where $\Delta^n(\mathbf{y}^*) = \mathbf{y}_{t_n} - \mathbf{y}_{t^*}$ denotes the physical evolution over the interval $[t^*, t_n]$. *The cumulative intervention error $\epsilon_{int}(n)$ is bounded as:*

$$\epsilon_{int}(n) \leq \epsilon_{stat} + \epsilon_{dyn}(t_n - t^*). \qquad (2.8)$$

**Implication.** Thm. 2.1 establishes a concrete link between structural alignment and interventional causal consistency. If static fidelity and dynamic coherence (Prop. 2.1) hold with uniform bounds, then interventions applied directly in the physical state space admit latent rollouts whose decoded trajectories remain physically plausible over multiple steps. Importantly, this guarantee concerns *intervention-consistent* rollouts under physically realizable perturbations of the system state. Moreover, this result does *not* require coordinate-wise disentanglement of the latent representation. Instead, it relies solely on the unique identifiability of the physical system up to a linear reparameterization. Accordingly, *distributed representations* that preserve the underlying system structure are consistent with supporting reliable reasoning under intervention. In this sense, structural alignment provides a minimal and testable set of necessary conditions for world models intended for scientific use.

*Remark* 2.3 (***On Counterfactual Reasoning***). Counterfactual reasoning is often regarded as the strongest level in causal hierarchies (Pearl, 2009). This work deliberately does *not* claim counterfactual reasoning, but rather focuses on *interventional* queries on physically realizable states. In scientific domains, counterfactual states are rarely observable and often ill-defined (see p. 220, Pearl (2009)), whereas interventions on physical variables are meaningful and empirically grounded. Our framework therefore targets a level of causal reasoning that is both testable and operational in real-world scientific systems.

## 3. The TC-Bench Dataset

Tropical cyclones provide a scientifically rich test case for studying learned representations: their evolution is governed by a small set of well-established physical variables, yet these variables are only indirectly observable through high-dimensional satellite imagery. We introduce TC-Bench, a *versioned, global* tropical cyclone benchmark dataset with a *unified and fully reproducible construction pipeline*. Unlike prior benchmarks that require manual curation or periodic maintenance (Knapp & Kossin, 2007; Knapp, 2008b), TC-Bench enables researchers to automatically reconstruct up-to-date global cyclone datasets across all basins, substantially extending existing resources in both lifecycle coverage and geographic diversity (Kitamoto et al., 2023; 2024).

The version of TC-Bench used in this study spans global tropical cyclones from 1980 to 2024 and integrates two complementary data sources: (1) the International Best Track Archive for Climate Stewardship (IBTrACS

v4r01) (Knapp et al., 2010; Gahtan et al., 2024), which provides storm metadata including minimum central pressure, maximum sustained wind, and storm location across multiple reporting agencies; and (2) geostationary infrared brightness temperature imagery from GridSat-B1 (Knapp et al., 2014), from which we extract fixed-size crops centered at reported storm locations. As of 2024, this pipeline yields 3,928 raw tropical cyclones across seven agencies, yielding a curated benchmark dataset with truly global basin coverage. Construction details, preprocessing, and summary statistics are reported in App. D.

Within the scope of this paper, TC-Bench serves as a controlled evaluation substrate for probing *scientific alignment*. Following the Digital Typhoon convention (Kitamoto et al., 2023), the observation space $\mathcal{X}$ consists of two-dimensional infrared satellite images, while the physical state space $\mathcal{Y}$ is defined by minimum central pressure ($P_c$) and maximum sustained wind speed ($V_m$). Because wind estimates are subject to substantial reporting heterogeneity across agencies (e.g., 1-minute vs. 10 minute averaging), our primary analyses focus on $P_c$, with complementary results involving $V_m$ reported in App. E.

## 4. Probing for Scientific Alignment

This section introduces a structured probing framework for assessing *scientific alignment* in learned representations, following the hierarchy of scientific queries summarized in Tab. 1. We evaluate alignment progressively along three dimensions. First, we test *Static Fidelity* and *Dynamic Coherence*, which assess whether physical state variables (minimum central pressure $P_c$) and their temporal evolution are linearly identifiable from the latent representation. Second, we examine *Manifold Consistency* ($Q_{con}$), asking whether independently recovered physical variables ($P_c, V_m$) jointly respect known physical couplings. Finally, we analyze the failure modes revealed by these probes to identify systematic breakdowns of alignment across intensity regimes. Together, these analyses provide a principled diagnostic of when and how visually strong representations fail to support physically meaningful reasoning.

### 4.1. Experimental Setup

We first specify the shared dataset partition, model suite, and probing protocol used across all three alignment probes.

**Dataset and regime definition.** To evaluate the uniformity requirement introduced in Defn. 2.3, we partition the dataset into intensity-based regimes. While maximum sustained wind $V_m$ is retained for completeness, regimes are defined using minimum central pressure $P_c$ to avoid inconsistencies arising from agency-specific wind reporting. We define the intense regime as $P_c < 980$ hPa and the moderate regime

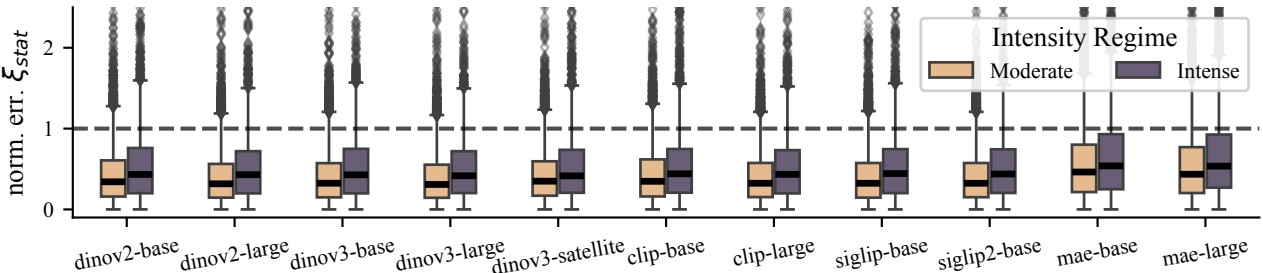

*Figure 2. Static Fidelity ($Q_{stat}$) and Regime Degradation*: Normalized absolute error ($\xi_{stat}$) of state estimation across diverse VFM architectures, stratified by intensity. A consistent performance gap is observed across all model families: the Intense regime exhibits significantly higher median errors and variance compared to the Moderate regime. This confirms the Visual Saturation hypothesis, where representations struggle to resolve fine-grained intensity differences in the high-intensity tail due to feature saturation.

as $P_c \geq 980$ hPa. This threshold corresponds to the point at which eye-like structures become statistically prevalent and visually dominant in infrared imagery (Dvorak, 1975).

**Models and probing protocol.** We evaluate a diverse suite of frozen vision foundation models (VFMs), including general-purpose vision models (DINOv2 (Oquab et al., 2023), DINOv3 (Siméoni et al., 2025), CLIP (Radford et al., 2021), SigLIP (Zhai et al., 2023), SigLIP2 (Tschannen et al., 2025)) and generative architectures such as MAE (He et al., 2022). For each model, we extract a global representational summary from the frozen backbone. Since our objective is single-scalar regression of minimum central pressure $P_c$ rather than dense spatial prediction, we use the CLS token as the primary representation. A comparative analysis using the aggregated spatial mean $\mathbf{z}_{sp}$ is provided in App. E. All representations are evaluated using a restricted linear probe $L \in \mathcal{H}_{lin}$. This design choice directly reflects our definition of structural alignment: successful prediction under a linear readout implies that the physical variable is *linearly identifiable* within the latent space, rather than recovered through decoder expressivity (Tab. 1).

**Controlling for alternative explanations.** Differences in model behavior across intensity regimes could in principle arise from confounding factors unrelated to representation quality, including sample imbalance, trivial correlations in the data, limited probe capacity, or biases specific to static image pretraining. To isolate representational effects, all analyses in this section are conducted on *regime-balanced subsets* with a unified linear probing protocol.

Beyond this primary control, we provide additional sanity checks in App. E: nonlinear probes (MLP and transformer probe) rule out a purely linear-probe artifact in App. E.2; task-specific models trained from scratch verify that pressure remains statistically recoverable from the imagery in App. E.1; spatial-token aggregation rules out dependence on the CLS token in App. E.3; and video-based models such as VideoMAE (Tong et al., 2022), V-JEPA2 (Assran et al.,

2025) and X-CLIP (Ni et al., 2022) with temporal predictive or video-language objectives test whether the failure is specific to static image pretraining (App. E.4).

> **Level 1: Static Fidelity ($Q_{stat}$)**
>
> **Probe:** Can the model linearly identify the physical state $\mathbf{y} = P$ from a latent representation $\mathbf{z} = g(\mathbf{x})$ extracted from a static frame $\mathbf{x} \in \mathcal{X}$?

**Probing protocol.** We adopt a trajectory-level split to prevent temporal or spatial leakage between training and evaluation. A linear surrogate $h \in \mathcal{H}_{lin}$ is trained on $\mathcal{M}_{train}$ following the regime-balanced evaluation protocol described in § 4.1. We report the normalized static fidelity error

$$\xi_{stat} := \frac{\|h(\mathbf{z}) - P_c\|}{\sigma(P_c)} \quad (4.1)$$

defined as the pointwise absolute error normalized by the global standard deviation of $P_c$. Under this metric, $\xi_{stat} = 1.0$ corresponds to the performance of a naive mean estimator.

**Results.** Across all evaluated Vision Foundation Models (VFMs), we observe a systematic performance degradation in the intense regime (see Fig. 2). While errors remain stable and tightly concentrated in the Moderate regime, both the median error and the frequency of catastrophic outliers ($\epsilon_{stat} > 1.0$) increase sharply once the 980 hPa threshold is crossed. Notably, this behavior is consistent across model families, indicating a regime-dependent failure of linear identifiability rather than a model-specific artifact. Furthermore, we show in App. E.2 that this degradation is not removed by increasing probe capacity: nonlinear MLP and Transformer probes exhibit the same qualitative moderate-to-intense gap, while the geometric diagnostics in § 4.2 show that the underlying feature distribution itself contracts in the intense regime.

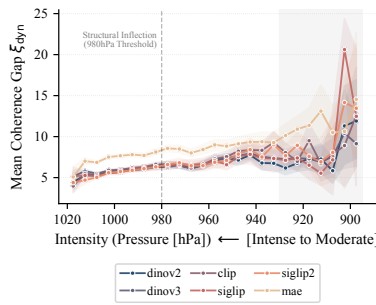

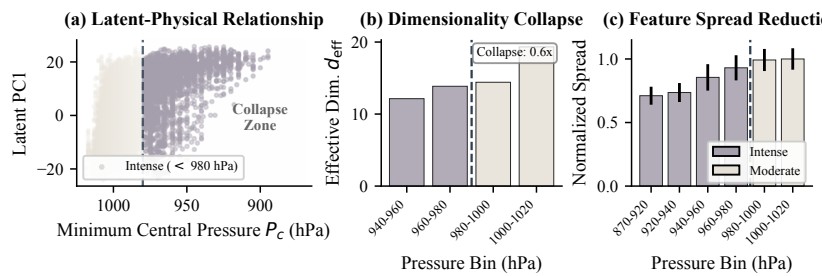

*Figure 3. Dynamic Coherence Decay ($Q_{dyn}$) across model families. The gap $\xi_{\mathrm{dyn}}$ increases with intensity ($P_c \downarrow$), with error spikes in the catastrophic regimes ($P_c < 920$ hPa), indicating degraded temporal alignment in extremes and compounding dynamical miscalibration.*

*Figure 4. Failure Mode Analysis:* DINOv3-base representations collapse in intense regimes. (a) PC1 varies systematically with pressure in moderate regimes, but this variation compresses in intense regimes, indicating reduced latent resolution of physically meaningful intensity differences. (b) Effective dimensionality drops in intense pressure bins, showing that within-bin variation concentrates into fewer latent directions. (c) Mean pairwise feature spread decreases in the same regime, showing that physically distinct storms are mapped closer together in latent space.

---

### Level 2: Dynamic Coherence ($Q_{\mathrm{dyn}}$)

**Probe:** Given the optimal readout $L$ from $Q_{\mathrm{stat}}$, does the projected latent displacement $L\Delta\mathbf{z}_t$ align with the corresponding physical state evolution $\Delta\mathbf{y}_t$?

---

**Probing Protocol.** We follow the same trajectory-level split and regime-balanced evaluation protocol as in $Q_{\mathrm{stat}}$. Dynamic coherence is evaluated via finite differences, exploiting the fixed temporal resolution of the data ($\Delta t = 3$ hours). Specifically, we measure the misalignment

$$\xi_{\mathrm{dyn}} := \|L\Delta\mathbf{z}_t - \Delta\mathbf{y}_t\| \tag{4.2}$$

which approximates the gap between latent and physical time derivatives up to a constant scaling.

**Results.** Fig. 3 reports the dynamic coherence gap $\epsilon_{\mathrm{dyn}}$ as a function of cyclone intensity, indexed by minimum central pressure $P_c$. Consistent with the static probe, $\epsilon_{\mathrm{dyn}}$ remains stable in Moderate regimes but increases sharply once the 980 hPa threshold is crossed. This degradation compounds with increasing variance and culminates in a pronounced error spike in the catastrophic regimes ($P_c < 920$ hPa), indicating a breakdown of latent–physical dynamical alignment.

---

### Level 3: Manifold Consistency ($Q_{\mathrm{con}}$)

**Probe:** Do the linearly recovered state variables $(\widehat{P}_c, \widehat{V}_m)$ respect known physical couplings between pressure and wind in tropical cyclones?

---

**Probing Protocol.** To assess *manifold consistency*, we test whether independently recovered pressure and wind estimates obey a basic physical coupling observed in tropical cyclones. While no closed-form law fully characterizes the

pressure–wind relationship, gradient wind balance implies that, all else being equal, lower-latitude storms require higher wind speeds to sustain the same central pressure deficit due to the reduced Coriolis force ($f \propto \sin\phi$). This yields a simple, testable monotonic constraint: for storms with comparable minimum central pressure, the expected maximum wind speed at low latitudes ($|\phi| < 15°$) should exceed that at higher latitudes ($|\phi| > 25°$). We quantify the ground-truth separation as

$$\Delta V_m := V_m^{\text{low-lat}} - V_m^{\text{high-lat}} \tag{4.3}$$

where $V_m$ denotes the 1-minute maximum sustained wind, and evaluate whether the recovered winds $\Delta\hat{V}_m$ exhibit the same ordering. Manifold inconsistency is measured by the relative deviation

$$\psi_{\mathrm{con}} := \frac{\|\Delta V_m - \Delta\hat{V}_m\|}{\Delta V_m} \tag{4.4}$$

with smaller values indicating better preservation of the pressure–wind manifold structure.

**Results.** Across models, manifold consistency deteriorates substantially with increasing cyclone intensity. In the Moderate regime ($P_c \geq 980$ hPa), the constraint violation remains limited, with $\psi_{\mathrm{con}} \approx 20\%$. In contrast, in the Intense regime ($P_c < 980$ hPa), $\psi_{\mathrm{con}}$ increases sharply to approximately $55\%$, indicating a failure to preserve physically meaningful pressure–wind structure under stronger dynamical forcing. We provide further qualitative results on this manifold misalignment in App. E.

#### 4.2. Failure Mode Analysis

To identify the source of the regime-dependent failures observed in the structural alignment probes, we analyze the *intrinsic geometry* of the latent representations produced by DINOv3-base, the strongest-performing model under

our linear evaluation protocol. All analyses are performed within pressure bins containing sufficiently many samples ($N \geq 500$).

**Experimental protocol.** We consider three complementary diagnostics of latent geometry. First, in Fig. 4a, we apply PCA to the frozen representations and examine the relationship between the first principal component (PC1) and minimum central pressure $P_c$. This tests whether the dominant latent direction preserves physically meaningful variation in cyclone intensity. Importantly, the sign or absolute value of PC1 has no physical meaning by itself; PCA directions are identifiable only up to sign. The relevant quantity is not whether intense storms have high or low PC1 values, but whether physically meaningful variation in $P_c$ remains resolved along dominant latent directions. Fig. 4 (a) shows that this resolution diminishes in the intense regime. Second, in Fig. 4 (b), we compute the *effective dimensionality* of the within-bin feature covariance spectrum using the participation ratio:

$$d_{\text{eff}} = \frac{\left(\sum_i \lambda_i\right)^2}{\sum_i \lambda_i^2}, \qquad (4.5)$$

where $\lambda_i$ are the eigenvalues of the within-bin covariance matrix. This quantity estimates how many orthogonal directions contribute substantially to feature variation: high effective dimensionality indicates that variance is distributed across many latent directions, whereas low effective dimensionality indicates concentration into a few dominant directions. Third, in Fig. 4 (c), we measure the *feature spread*, defined as the mean pairwise Euclidean distance between centered features within each pressure bin. This quantifies how locally dispersed the representations remain in feature space; lower spread indicates that physically distinct samples are mapped closer together.

**Results.** As shown in Fig. 4, DINOv3 representations undergo a systematic collapse in intense regimes. Once the minimum central pressure falls below $980\,\text{hPa}$, physically distinct cyclones are mapped to a narrower region of latent space, reflected by the shrinking variation of PC1 with pressure in Fig. 4a. This collapse is accompanied by a sharp reduction in effective dimensionality, by approximately $60\%$ in Fig. 4b, and a corresponding decrease in feature spread in Fig. 4c. Together, these diagnostics indicate that visually extreme but physically distinct storms are compressed into a lower-rank latent subspace, causing a loss of distinguishable physical degrees of freedom.

**Interpretation.** Inferring pressure from satellite imagery is intrinsically more difficult in the high-intensity regime, since visual differences between storms become subtler and measurement noise increases. However, this difficulty alone does not explain the behavior of frozen VFM representations. A supervised pixel-space baseline trained from scratch achieves consistently lower error in the same regime, indicating that pressure-related signal remains statistically present in the observations (Fig. 6). In contrast, frozen VFM representations exhibit reduced effective dimensionality and diminished feature spread precisely where physical variation remains important (Fig. 4). These results therefore point to a systematic misalignment between the learned latent geometry and the physically relevant structure, rather than to task difficulty alone.

# 5. Related Works

**World Models and OOD Generalization: When Passing the Test Is Not Enough.** Recent vision foundation models (e.g., VideoMAE (Wang et al., 2023; Tong et al., 2022; He et al., 2022), V-JEPA (Bardes et al., 2024; Assran et al., 2025), DINO (Oquab et al., 2023; Siméoni et al., 2025), SigLIP (Zhai et al., 2023; Tschannen et al., 2025), CLIP (Radford et al., 2021)) are often motivated as general-purpose world models, aiming to learn representations that remain predictive under distribution shift. We focus exclusively on perception-based world models, where the physical state must be inferred from unstructured visual inputs, in contrast to emulator-based models operating on fully observed physical grids (Price et al., 2025; Alet et al., 2025; Bonev et al., 2025; Bi et al., 2023; Pathak et al., 2022). In this paradigm, success is typically measured by perceptual or predictive robustness—e.g., reconstruction quality, semantic consistency, or downstream accuracy on OOD benchmarks. However, such evaluations admit a blind spot: a model may generalize perceptually under distribution shift while failing to preserve physically meaningful degrees of freedom. We show that even models performing well on standard OOD benchmarks, including those targeting geographical shift, can collapse distinct physical states into a narrow latent region (Fig. 1). Thus, passing perceptual OOD tests does not certify that a representation functions as a valid surrogate for physical state variables.

**Probing and Representation Diagnostics: Beyond Average-Case Recoverability.** Probing is the standard tool for interpreting frozen representations, commonly implemented via linear classifiers or regressors (Alain & Bengio, 2016; Hewitt & Manning, 2019). In scientific ML, probes are often used to assess whether representations encode physical quantities or align with scientific concepts (Donhauser et al., 2025). Existing critiques primarily emphasize over-interpretation, noting that probes may succeed by exploiting superficial correlations rather than faithful structure (Elazar et al., 2021). We identify a more consequential failure mode: *regime-dependent collapse*. This failure mode is invisible to standard metrics and reflects a breakdown of latent geometry. By introducing alignment probing as a test of necessary structural con-

ditions (Tab. 1), our framework exposes a failure mode missed by conventional probe-based evaluation.

**Failure Modes of Scientific Representations Under Perceptual Saturation.** Beyond input modality and evaluation protocol, an open question remains: *what concrete failure modes allow perceptually robust world models to violate scientific structure?* We identify a previously under-characterized mechanism: *regime-dependent latent collapse induced by visual saturation*. In intense regimes, visually similar observations may correspond to sharply different physical states, allowing predictive success to mask representational failure (Geirhos et al., 2020). We show that representations can remain predictive and OOD-robust while collapsing along physically meaningful axes, effectively compressing distinct states into a low-dimensional latent region (Fig. 1). Crucially, this collapse does not manifest as degraded average performance or perceptual inconsistency, consistent with prior observations that standard metrics can obscure internal representational pathologies (Elazar et al., 2021). Instead, it compromises the ability to support coherent dynamics, regime transitions, and constraint satisfaction—properties required for scientific reasoning and surrogate physical modeling (Cummins, 1989; Swoyer, 1991).

**Relation to Causal Representation Learning.** Causal representation learning (CRL Schölkopf et al. (2021)) aims to recover latent variables corresponding to causal factors, often under strong assumptions about interventions or multi-environment diversity (von Kügelgen et al., 2024; Zhang et al., 2024a;b; Yao et al., 2023; 2024). Such assumptions are rarely satisfied in large-scale observational settings, particularly when physical states are only indirectly observed via satellite imagery. Our framework offers a more operational alternative: rather than seeking element-wise causal identifiability, we test for *Structural Isomorphism* (Defn. 2.3), requiring that representations preserve physical coordinates up to a distributed linear reparameterization. This condition theoretically guarantees intervention-consistent rollouts (Thm. 2.1) while remaining empirically testable on foundation models. As such, our framework bridges predictive world models and causal discovery, focusing on a regime that is both scientifically meaningful and empirically accessible.

## 6. Conclusion and Limitations

This work identifies a *Perception–Physics Paradox* in vision foundation models (VFMs) applied to tropical cyclone satellite imagery: models can appear perceptually robust while failing to preserve physically meaningful latent structure. In intense regimes, where visual saturation such as eye formation coincides with subtle but important physical variation, frozen VFM representations lose resolution along critical physical dimensions (§ 4.2). This failure is not revealed by

average-case metrics or visually defined OOD tests alone, such as the cross-agency evaluation in Fig. 1(c).

To study this gap, we introduce *Scientific Alignment* and operationalize one testable aspect through *structural isomorphism* (Defn. 2.3). This yields locally falsifiable necessary conditions—static fidelity, dynamic coherence, and manifold consistency—which we evaluate through a systematic probing pipeline (Tab. 1). Together with TC-BENCH, a reproducible tropical cyclone benchmark dataset designed for physically targeted stress tests, these probes show consistent structural misalignment of existing VFMs in physically extreme regimes (§ 4).

Notably, the *Perception–Physics Paradox* is not specific to tropical cyclones. It can arise whenever observations become locally insensitive to physically meaningful variation in extreme regimes, even though the underlying signal remains present and recoverable. Similar saturation mechanisms appear in other high-impact remote sensing settings, such as wildfire monitoring, where thermal observations can saturate once flames fill a sensor pixel (Wooster et al., 2005, Sec. 3.2). These settings are often rare, destructive, and difficult to instrument, which motivates benchmarks that explicitly target extreme-regime coverage.

More broadly, our framework provides necessary rather than sufficient conditions for scientific usability. Future work should extend these probes to other physical systems and develop alignment-aware objectives based on temporal consistency, physical constraints, and multi-view or multi-modal supervision.

## Impact Statement

This work contributes TC-BENCH, a global and reproducible tropical cyclone dataset construction pipeline, and a diagnostic framework for evaluating scientific alignment in learned representations. TC-BENCH unifies multi-agency best-track records with satellite observations, enabling transparent reconstruction and auditing as source data evolve. We hope this supports careful evaluation, reproducibility, and principled representation learning for high-stakes scientific applications.

### Acknowledgements

We are especially grateful to Michael Tschannen for his generous support, careful feedback, and helpful suggestions, and to the CLAI group at ISTA for valuable discussions throughout the project. Dingling Yao acknowledges support from a Google Initiated Gift and from the Google PhD Fellowship Program, with support from Google.org. Adeel Pervez was supported by funding from the European Union's Horizon 2020 research and innovation programme under the Marie Skłodowska-Curie Grant Agreement No. 101034413.

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

# Appendix

## A. Notation and Terminology

This section provides a glossary of symbols and notations used throughout the paper.

| | |
|---|---|
| $\mathcal{P}$ | Constraint Set |
| $\mathcal{S}$ | Physical System |
| $\mathcal{X}$ | Observation Space |
| $\mathcal{Y}$ | Physical State Space |
| $\phi$ | Observation Function |
| $\mathbf{x}$ | Observation |
| $\mathbf{y}$ | Physical State |
| $\mathcal{Z}$ | Latent Space |
| $\mathbf{z}$ | Latent Representation |
| $g$ | Encoder |
| $\mathcal{M}$ | Regime Family |
| $\mu$ | Regime Measure |
| $\mathcal{D}_\mu$ | Data distribution induced by $\mu$ |
| $f_\mu$ | Regime-specific Vector Field |
| $Q$ | Structural Alignment Probe |
| $\mathcal{H}$ | Surrogate Model Class |
| $\mathcal{R}$ | Reference Domain |
| $\psi$ | Probe Evaluation Metrics |
| $h$ | Surrogate Model |
| $\epsilon_{\text{stat, dyn, con}}$ | Theoretical Error Bounds for Static Fidelity, Dynamic Coherence and Manifold Consistency |
| $\xi_{\text{stat, dyn, con}}$ | Estimated Errors of Static Fidelity, Dynamic Coherence and Manifold Consistency Probes |
| $P_c$ | Minimum Central Pressure |
| $V_m$ | Maximum Sustained Wind |

## B. Proofs

This section summarizes the proofs for theoretical statements in § 2.

**Proposition 2.1** (Uniform Implications of Structural Isomorphism). *Given a physical system $\mathcal{S}$ with bounded vector field $\|\mathbf{f}_\mu(\mathbf{y})\| \leq K$ for all regime $\mu \in \mathcal{M}$, if a representation $\mathbf{z} = g(\mathbf{x})$ is structurally isomorphic to $\mathcal{S}$ (Defn. 2.3) and the constraints $\mathcal{P}$ in $\mathcal{S}$ are $\Lambda_\mathcal{P}$-Lipschitz continuous, then there exists a linear decoder $L \in \mathbb{R}^{m \times d}$ (e.g. a left inverse of $\mathbf{A}$) such that, uniformly over the regime family $\mathcal{M}$, the following hierarchical properties hold:*

*(i) Static Fidelity. The physical state is uniformly recoverable from the representation:*

$$\sup_{\mu \in \mathcal{M}} \mathbb{E}_{\mathbf{y} \sim \mathcal{D}_\mu}\left[\|\mathbf{y} - L\mathbf{z}\|\right] \leq \|L\|\bar{\epsilon} =: \epsilon_{\text{stat}}. \tag{2.3}$$

*(ii) Dynamic Coherence. The decoded latent derivatives $\dot{\mathbf{z}}$ uniformly approximate the physical derivatives $\dot{\mathbf{y}}$:*

$$\sup_{\mu \in \mathcal{M}} \mathbb{E}_{(\mathbf{y}, \dot{\mathbf{y}}) \sim \mathcal{D}_\mu} \|\dot{\mathbf{y}} - L\dot{\mathbf{z}}\| \leq \|L\|\bar{\delta}K =: \epsilon_{\mathrm{dyn}}, \tag{2.4}$$

*where $L$ is the same decoder as in (i).*

*(iii) Manifold Consistency. The linear reconstruction approximately satisfies the system invariants $\mathcal{P}$:*

$$\sup_{\mu \in \mathcal{M}} \mathbb{E}_{\mathbf{y} \sim \mathcal{D}_\mu} \big[\|\mathcal{P}(L\mathbf{z})\|\big] \leq \Lambda_\mathcal{P}\|L\|\bar{\epsilon} =: \epsilon_{\mathrm{con}}. \tag{2.5}$$

*Proof.* To prove Prop. 2.1, we demonstrate that the definition of Structural Isomorphism (Defn. 2.3) establishes a rigid linear bridge between the latent space $\mathcal{Z}$ and the physical state $\mathcal{Y}$. This bridge ensures that the properties of the physical system are preserved in the latent manifold, up to a bounded error.

**Linear Decoder $L$:** Since $\mathbf{A} : \mathcal{Y} \to \mathcal{Z}$ is an injective linear embedding with $\dim(\mathcal{Z}) = k \geq \dim(\mathcal{Y}) = m$, there exists a Moore-Penrose pseudoinverse (or left-inverse) $L \in \mathbb{R}^{m \times k}$ such that:

$$L\mathbf{A} = \mathbf{I}_m, \tag{B.1}$$

where $\mathbf{I}_m$ is the $m \times m$ identity matrix. This $L$ acts as our linear decoder.

**(i) Proof of Static Fidelity.** By Defn. 2.3, we have $\mathbf{z} = \mathbf{A}\mathbf{y} + \epsilon_\mu(\mathbf{y})$. Multiplying both sides by $L$:

$$L\mathbf{z} = L\mathbf{A}\mathbf{y} + L\epsilon_\mu(\mathbf{y}) = \mathbf{y} + L\epsilon_\mu(\mathbf{y}). \tag{B.2}$$

The error in recoverability is $\|\mathbf{y} - L\mathbf{z}\| = \|L\epsilon_\mu(\mathbf{y})\|$. Taking the supremum over the regime family $\mathcal{M}$:

$$\sup_{\mu \in \mathcal{M}} \mathbb{E}_{\mathbf{y} \sim \mathcal{D}_\mu}[\|\mathbf{y} - L\mathbf{z}\|] \leq \|L\| \cdot \sup_{\mu \in \mathcal{M}} \mathbb{E}_{\mathbf{y} \sim \mathcal{D}_\mu}[\|\epsilon_\mu(\mathbf{y})\|] \leq \|L\|\bar{\epsilon} := \epsilon_{\mathrm{stat}}. \tag{B.3}$$

**(ii) Proof of Dynamic Coherence.** Differentiating the isomorphism relation $\mathbf{z} = \mathbf{A}\mathbf{y} + \epsilon_\mu(\mathbf{y})$ with respect to time, and applying the chain rule to the residual term $\dot{\epsilon}_\mu(\mathbf{y}) = J_{\epsilon_\mu}(\mathbf{y})\dot{\mathbf{y}}$:

$$\dot{\mathbf{z}} = \mathbf{A}\dot{\mathbf{y}} + J_{\epsilon_\mu}(\mathbf{y})\dot{\mathbf{y}}. \tag{B.4}$$

Multiplying by the decoder $L$ (where $L\mathbf{A} = \mathbf{I}$):

$$L\dot{\mathbf{z}} = \dot{\mathbf{y}} + LJ_{\epsilon_\mu}(\mathbf{y})\dot{\mathbf{y}}. \tag{B.5}$$

The gap between the decoded latent transition and the vector field is:

$$\|\dot{\mathbf{y}} - L\dot{\mathbf{z}}\| = \|LJ_{\epsilon_\mu}(\mathbf{y})\dot{\mathbf{y}}\| \leq \|L\| \cdot \|J_{\epsilon_\mu}(\mathbf{y})\| \cdot \|\dot{\mathbf{y}}\|. \tag{B.6}$$

Using the assumption that the vector field is bounded ($\|\dot{\mathbf{y}}\| \leq K$), we take the expectation and supremum:

$$\begin{aligned} \sup_{\mu \in \mathcal{M}} \mathbb{E}[\|\dot{\mathbf{y}} - L\dot{\mathbf{z}}\|] &\leq \|L\|K \cdot \sup_{\mu \in \mathcal{M}} \mathbb{E}[\|J_{\epsilon_\mu}(\mathbf{y})\|] \\ &\leq \|L\|K\bar{\delta} := \epsilon_{\mathrm{dyn}}. \end{aligned} \tag{B.7}$$

**(iii) Proof of Manifold Consistency.** For any invariant constraint $\mathcal{P}(\mathbf{y}) = 0$, we evaluate the decoded state $L\mathbf{z} = \mathbf{y} + L\epsilon_\mu(\mathbf{y})$. Since $\mathcal{P}$ is Lipschitz continuous with constant $\Lambda_\mathcal{P}$, and $\mathcal{P}(\mathbf{y}) = 0$:

$$\|\mathcal{P}(L\mathbf{z})\| = \|\mathcal{P}(\mathbf{y} + L\epsilon_\mu(\mathbf{y})) - \mathcal{P}(\mathbf{y})\| \leq \Lambda_\mathcal{P}\|L\epsilon_\mu(\mathbf{y})\|. \tag{B.8}$$

The uniform bound across all regimes is:

$$\sup_{\mu \in \mathcal{M}} \mathbb{E}_{\mathbf{y}}[\|\mathcal{P}(L\mathbf{z})\|] \leq \Lambda_\mathcal{P}\|L\| \sup_{\mu \in \mathcal{M}} \mathbb{E}_{\mathbf{y}}[\|\epsilon_\mu(\mathbf{y})\|] = \Lambda_\mathcal{P}\|L\|\bar{\epsilon} := \epsilon_{\mathrm{con}}. \tag{B.9}$$

This confirms that the latent representation remains "physically plausible" for all regimes $\mu \in \mathcal{M}$. $\square$

**Theorem 2.1** (Interventional Grounding). *Let $\mathcal{Y}$ be a compact manifold and assume a bounded physical vector field $\|\mathbf{f}_\mu\| \leq K$ for all $\mu \in \mathcal{M}$. Let $g : \mathcal{X} \to \mathcal{Z}$ be an encoder that satisfies the structural alignment hierarchy (Prop. 2.1) with uniform residuals $\{\epsilon_{stat}, \epsilon_{dyn}\}$ under a linear decoder $L$. Let $G = g \circ \phi$ denote the induced mapping from the state space $\mathcal{Y}$ to the latent space $\mathcal{Z}$. Then, for a hard intervention $do(\mathbf{y}^*)$ applied at time $t^*$, the $n$-step interventional rollout satisfies*

$$\|L\Delta^n G(\mathbf{y}^*) - \Delta^n(\mathbf{y}^*)\| \leq \epsilon_{int}(n), \tag{2.7}$$

*where $\Delta^n(\mathbf{y}^*) = \mathbf{y}_{t_n} - \mathbf{y}_{t^*}$ denotes the physical evolution over the interval $[t^*, t_n]$. The cumulative intervention error $\epsilon_{int}(n)$ is bounded as:*

$$\epsilon_{int}(n) \leq \epsilon_{stat} + \epsilon_{dyn}(t_n - t^*). \tag{2.8}$$

*Proof.* The total error of the interventional rollout can be decomposed into the error in the initial state reconstruction (static) and the error accumulated during trajectory integration (dynamic).

(I) **Initial Mapping (Base Case).** At the moment of intervention $t^*$, the physical state is forced to $\mathbf{y}^*$. By Definition 2.3 and Proposition 2.1(i), the latent representation $\mathbf{z}^*$ corresponds to $\mathbf{y}^*$ up to the static fidelity bound. Thus, the decoded initial state $L\mathbf{z}^*$ satisfies:

$$\|L\mathbf{z}^* - \mathbf{y}^*\| \leq \epsilon_{\text{stat}}. \tag{B.10}$$

(II) **Compounding Dynamic Error.** Consider the evolution over the interval $[t^*, t_n]$. The physical displacement is the integral of the vector field:

$$\Delta^n(\mathbf{y}^*) = \int_{t^*}^{t_n} \dot{\mathbf{y}}_\tau \, d\tau. \tag{B.11}$$

Similarly, the displacement in the decoded latent space is the integral of the decoded derivatives:

$$L\Delta^n G(\mathbf{y}^*) = \int_{t^*}^{t_n} L\dot{\mathbf{z}}_\tau \, d\tau. \tag{B.12}$$

From Proposition 2.1(ii) (Dynamic Coherence), we know that uniformly over the trajectory:

$$\|L\dot{\mathbf{z}} - \dot{\mathbf{y}}\| \leq \epsilon_{\text{dyn}}. \tag{B.13}$$

We verify the bound for the trajectory difference using the triangle inequality for integrals:

$$\begin{aligned}
\|L\Delta^n G(\mathbf{y}^*) - \Delta^n(\mathbf{y}^*)\| &= \left\| \int_{t^*}^{t_n} (L\dot{\mathbf{z}}_\tau - \dot{\mathbf{y}}_\tau) \, d\tau \right\| \\
&\leq \int_{t^*}^{t_n} \|L\dot{\mathbf{z}}_\tau - \dot{\mathbf{y}}_\tau\| \, d\tau \\
&\leq \int_{t^*}^{t_n} \epsilon_{\text{dyn}} \, d\tau \\
&= \epsilon_{\text{dyn}}(t_n - t^*).
\end{aligned} \tag{B.14}$$

(III) **Total Interventional Error.** The total error in the rolled-out state $\hat{\mathbf{y}}_{t_n}$ relative to the true counterfactual $\mathbf{y}_{t_n}$ is the sum of the initial displacement error and the integration drift. Note that technically $\Delta^n$ measures displacement relative to the initial condition. Combining (I) and (II), the bound on the absolute state error is:

$$\epsilon_{\text{int}}(n) \leq \underbrace{\epsilon_{\text{stat}}}_{\text{Initial Condition}} + \underbrace{\epsilon_{\text{dyn}}(t_n - t^*)}_{\text{Integration Drift}}. \tag{B.15}$$

$\square$

## C. Physics of Tropical Cyclones

Tropical cyclones constitute a canonical example of a high-impact, multiscale physical system whose evolution is governed by a small number of interacting dynamical and thermodynamical principles, yet is only indirectly observable at scale. This section provides a concise overview of the physical mechanisms underlying tropical cyclone formation, structure, and intensification, emphasizing the force balances, energy constraints, and observable signatures that link latent physical state variables—such as minimum central pressure and maximum sustained wind—to satellite infrared imagery. We aim to summarize the key physical relationships and approximations that inform how cyclone intensity is inferred from observations and motivate the study of physically meaningful representations in data-driven models.

### C.1. Large-Scale Forcing and Cyclone Genesis

Tropical cyclones (TC) are intense, rotating storms that develop over the warm waters of the tropical oceans, which supply them with thermal energy and water vapor. Usually, the word "tropical cyclone" is used for the ones that achieve a maximum sustained wind of at least 118 km h$^{-1}$, with the weaker ones being called "tropical storms" (63-117 km h$^{-1}$) or tropical depressions (<63 km h$^{-1}$). Other names are used for tropical cyclones occurring in specific parts of the world; for example, "hurricanes" for those developing in the North Atlantic Ocean and "typhoons" for those in the Western Pacific. Although tropical cyclone is a relatively rare phenomenon (a few tens per year worldwide), it shows some of the highest ever recorded wind intensities (Stern et al., 2016) and rain accumulations (Prat & Nelson, 2016), as well as some of the highest death tolls and economic costs among all natural disasters (Emanuel, 2003).

Tropical cyclones are widely studied through a combination of numerical simulations and observational data analysis, but the physical mechanisms underlying their formation and intensification remain poorly understood.

Tropical cyclones arise from synoptic-scale ($\sim 1000$ km) negative anomalies in the large-scale air pressure field over the tropical oceans. A radial negative pressure anomaly induces a force pointing inward, which makes air converge toward the low pressure, but at the scale we are considering, the magnitude of the Coriolis force, given by the rotation of the Earth, is enough to partially deflect the converging air toward the right/left in the Northern/Southern hemisphere, starting to create a clockwise/counterclockwise vortex. Furthermore, the low-level converging wind enhances the transfer of heat and the evaporation from the ocean, which feeds energy to the core of the cyclone, further decreasing its inner pressure and contributing also to the destabilization the air column, leading to a reinforcement of deep convective ascent.

The dissipation of wind energy induced by friction in the lower levels of the atmosphere eventually balances the energy input from the ocean, leading to the establishment of a steady state: at this point, the dynamical picture of the cyclone includes two main air circulations: the primary one, that consists of the vortex in the horizontal plane, (a roughly axisymmetric one), and the so called secondary circulation in the vertical plane, consisting of a overturning cell in which air ascends close to the center of the cyclone (in the so called eyewall) and reaches very high altitude, near the so called tropopause. At the tropopause a stable layer of air starts, so ascending air stops moving upward and starts diverging outward. This divergence leads to a compensation inflow in the lower level, that closes the loop.

### C.2. Force Balance in the Primary Circulation

The primary circulation can be described as originated by a balance of three main forces: the pressure gradient force, directed radially and pointing inward, given the low pressure center, the Coriolis force, given by the rotation of the Earth, which points to the right/left of the tangential wind, and therefore radially and outward, and finally the centrifugal force given by the rotating flow in the cyclone itself, which is also directed radially and pointing outward. Formally, it can be described by the following equation (Willoughby, 1990):

$$\frac{V^2}{r} + fV = \frac{1}{\rho}\frac{\partial P}{\partial r}, \tag{C.1}$$

where $V$ denotes the tangential speed at the surface and $P_s$ the surface pressure (in hPa). $r$ is the distance from the center (i.e., Earth's radius, $r \approx 6,371$km), $f$ is the Coriolis parameter which depends only on the latitude, written as $f = 2\Omega \cdot \sin(\text{lat}) \approx 15 \cdot 10^{-5} \cdot \sin(\text{lat})\ ^{rad}/s$ ($\Omega$ is the rotation rate of the Earth), $\rho$ is the air density, taking value of roughly $1.2^{kg}/m^3$ as we are considering the surface.

## C.3. Secondary Circulation and Deep Convection

In addition to the primary horizontal circulation, tropical cyclones exhibit a secondary circulation in the vertical plane (Persing et al., 2013). Near the surface, air spirals inward toward the cyclone center, where it ascends rapidly in a narrow annulus known as the eyewall. This ascending motion transports heat, moisture, and angular momentum upward through the troposphere.

As the rising air approaches the tropopause, it encounters a statically stable layer that inhibits further vertical motion. The flow therefore turns outward, forming an upper-level anticyclonic outflow that diverges away from the storm center. Mass continuity requires a compensating inflow near the surface, thereby closing the overturning circulation.

Deep convection in the eyewall plays a central role in the storm's energetics and dynamics. The release of latent heat during condensation warms the atmospheric column, further reducing surface pressure and reinforcing the pressure gradient that drives the primary circulation.

## C.4. Tropical Cyclones as Heat Engines

One of the simplest and yet most meaningful models of a tropical cyclone is the Carnot engine model introduced by Emanuel (1986). A Carnot engine is an ideal system that absorbs thermal energy from a source at temperature $T_2$, transforms part of it into mechanical energy, and releases the remaining into a sink at temperature $T_1$ with $T_2 > T_1$. In a Carnot machine a fluid undergoes a cycle consisting of four thermodynamic transformations: a reversible isothermal expansion (temperature remains constant) , an adiabatic expansion (no heat exchange with the environment), an isothermal compression and a final adiabatic compression. The Carnot cycle can be proved to be the most efficient thermodynamic cycle possible, i.e. the with the highest mechanical energy output $W$ for a given thermal energy input $Q_{in}$ and the efficiency $\eta$ can be calculated with the following formula:

$$\eta := \frac{W}{Q_{in}} = 1 - \frac{T_1}{T_2} \tag{C.2}$$

A tropical cyclone can indeed be described as a machine that absorbs thermal energy from the ocean and transforms part of it into kinetic energy (under the form of wind energy) and releases the remaining in the upper troposphere. In particular, when air converges toward the center, it undergoes an expansion (because the pressure decreases going inward), while the temperature increases slightly, since it is constrained by the sea surface temperature. When it reaches the eyewall, it quickly ascends for many kilometers, undergoing a strong expansion, that can be considered adiabatic, since it is fast enough to avoid huge heat losses to the surrounding. Once air reaches the tropopause, it starts diverging outward, toward higher pressure (so it is a compression) and loosing energy toward the outerspace through infrared radiative emission. In this leg radiative cooling and adiabatic compression warming roughly balance each other, so it can be considered as an isothermal compression. Eventually air stops diverging and starts sinking back to the surface, warming through an ideal adiabatic compression and closing the cycle. In this view the ocean is the heat source and the tropopause the heat sink, so $T_2 = SST$ where $SST$ is the sea surface temperature, and $T_1 = T_{tr}$, where $T_{tr}$ is the temperature of the tropopause. Carnot formula of Eq.C.2 allows then to compute the maximum mechanical energy output rate for a given input, and therefore the maximum wind speed achievable, called the potential intensity of the cyclone ($V_p$). After some manipulations, the following equation follows (the $T_{tr}$ factor at the denominator, instead of $SST$ as Eq. C.2 would predict, is due to the fact that some of the mechanical energy is dissipated through friction into heat, that is reused by the system (Bister & Emanuel, 1998)):

$$V_p^2 = \alpha \frac{SST - T_{tr}}{T_{tr}} \tag{C.3}$$

Where $\alpha$ is a product of different factors that accounts for the efficiency of the energy transfer between the ocean and the cyclone. In reality, potential intensity is almost never reached by the cyclones, because of many reasons including heat losses, cooling of the ocean surface given by wind-induced mixing and landfall, but it still provides an interesting upper limit.

From Eq. C.3 it is clear that $V_p$ depends on both $SST$ and $T_{tr}$, which are captured by infrared imaging of tropical cyclones, suggesting the importance of looking at the pattern of deep convective clouds and clear sky areas to understand the intensification of the storm. A mature cyclone tends to show a central area of a few tens of km of radius with a high infrared emission (meaning that no deep convective cloud is there), called the *eye* of the cyclone, surrounded by a ring of very low infrared emission, where the deepest convection happens, called the eyewall. Further away from the cyclone center other concentric deep convective bands alternated with non-convective regions can be typically observed (Emanuel, 2003).

Furthermore, there exist some methods to obtain a numerical estimate of the cyclone intensity from the its infrared image: the most widely used is the so called Dvorak technique (Dvorak, 1975): this method uses the difference in temperature between the eye and the surrounding cloud eyewall (directly calculated from their infrared radiance) to infer the value of the minimum sea level pressure (MSLP), since a colder cloud top generally means a more intense deep convection and therefore a lower pressure at the center. Dvorak technique yields good estimation of MSLP, avoiding the necessity of conducting more difficult and/or expensive *in-situ* measurements, but the recent advances of machine learning, image-based regression tasks can provide a stronger alternative (Kitamoto et al., 2023).

## D. The TC-Bench Dataset

This appendix provides additional engineering details of the dataset construction pipeline for TC-Bench, together with summary statistics of key physical variables, including minimum central pressure $P_c$ and maximum sustained wind speed $V_m$, reported by different weather agencies.

### D.1. Dataset Resources

TC-Bench is constructed from the International Best Track Archive for Climate Stewardship (IBTrACS), version 4r01 (Knapp et al., 2010; Gahtan et al., 2024), one of the most comprehensive global tropical cyclone datasets available, spanning over 180 years (1840–present) and covering all major tropical basins worldwide. In this work, we restrict attention to the period 1980–2024, corresponding to the modern satellite era with consistent global infrared coverage.

IBTrACS provides cyclone center locations (latitude and longitude) at a nominal *3-hour* temporal resolution, together with estimates of minimum central pressure $P_c$ and maximum sustained wind speed $V_m$. Because cyclone tracking and intensity estimation are performed by different regional meteorological agencies, certain variables are not defined consistently across the archive. The primary inconsistency relevant to this study concerns maximum sustained wind speed: most agencies report 10-minute averaged winds at 10 m altitude, whereas the U.S. National Weather Service uses a 1-minute averaging window. To avoid introducing systematic biases, our primary analyses focus on minimum central pressure $P_c$, while wind speed is retained for auxiliary analyses for manifold consistency.

### D.2. Data Cleaning and Preprocessing

**Physical labels.** We partition the dataset into nine sub-datasets, each corresponding to a distinct reporting agency. A dedicated quality-control step is applied to both $P_c$ and $V_m$, as these variables are often reported at coarser temporal resolution than the cyclone track (commonly ∼6 hours), and are frequently missing during the earliest and latest stages of storm development.

For each cyclone track, we identify the first and last timesteps at which both wind speed and pressure measurements are available, and discard observations outside this valid interval. Within the retained window, we linearly interpolate missing values to obtain a uniform 3-hour temporal resolution aligned with the track data. This procedure preserves lifecycle continuity while avoiding extrapolation beyond physically supported measurements.

**Satellite imagery.** Infrared satellite observations are obtained from the Geostationary IR Channel Brightness Temperature GridSat-B1 dataset (Knapp et al., 2014), which provides global, 3-hourly measurements from 1980–2024 over latitudes 70°S–70°N and longitudes 180°W–180°E. The data are provided on a regular latitude–longitude grid using a Plate Carrée projection, with a spatial resolution of approximately 0.07° (corresponding to ∼8 km at the equator), and are distributed in NetCDF format. GridSat reports nadir-most top-of-atmosphere brightness temperature, a quantity proportional to infrared radiance, calibrated using multiple inter-satellite normalization protocols (Desormeaux et al., 1993; Knapp, 2008a) to reduce systematic bias and variance across platforms.

Physically, infrared brightness temperature reflects the temperature at the effective cloud-top emitting level: absorption and re-emission by water vapor, cloud liquid water, and ice particles cause higher-altitude, colder cloud tops to appear darker in infrared imagery. As a result, these measurements provide indirect but informative cues about deep convection and storm organization, both of which play a central role in tropical cyclone intensification (see App. C for further physical background). For each cyclone timestep, we extract a $224 \times 224$ spatial window centered on the reported storm location, corresponding to a square region of approximately 1,800 km on each side. Each extracted frame undergoes a quality check to identify non-physical values (defined by the valid brightness temperature range 140–375 K), including missing values represented as `NaN`. We compute a per-frame validity mask based on both the proportion and spatial distribution of

*Table 2.* TC-BENCH dataset statistics by reporting agency (after cleaning) .

| Agency | # Cyclones |
|---|---|
| ATCF | 7 |
| BoM | 235 |
| HURDAT (Atlantic) | 485 |
| HURDAT (East Pacific) | 448 |
| NADI | 39 |
| New Delhi | 286 |
| Réunion | 299 |
| Tokyo | 750 |
| Wellington | 52 |
| **Total** | **2,601** |

*Table 3.* Summary statistics of physical variables in TC-BENCH.

| Variable | Mean | Std | Min | Max |
|---|---|---|---|---|
| Pressure (hPa) | 988.71 | 20.19 | 872.0 | 1024.0 |
| Wind speed (kt) | 47.43 | 23.16 | 3.0 | 185.0 |
| Latitude (deg) | 10.06 | 19.30 | -60.5 | 62.0 |
| Longitude (deg) | 24.63 | 102.96 | -180.0 | 179.9 |

missing pixels. Frames with extensive corruption are discarded. For frames with only a small number of missing pixels in non-critical regions, we replace invalid values with a constant fill value of 200 K. This value lies near the center of the empirical brightness temperature distribution and introduces no additional spatial structure, ensuring that the imputation does not inject artificial texture or bias into the imagery while preserving downstream compatibility.

### D.3. Dataset Inspection

We summarize global statistics of the physical variables in Tab. 3 and further examine their distribution across reporting agencies in Fig. 5. In particular, we visualize per-agency histograms of minimum central pressure $P_c$ and maximum sustained wind speed $V_m$ to assess dataset coverage, inter-agency variability, and potential sources of measurement heterogeneity. This inspection highlights systematic differences arising from agency-specific reporting practices, most notably for wind speed, and motivates our emphasis on pressure-based analyses in the main text.

## E. Additional Experimental Details and Results

This section summarizes implementation details and additional experimental results. Specifically, to control for potential sources of performance degradation in extreme regimes and to isolate representational collapse from other confounding factors, we conduct the following sanity checks in Apps. E.1 to E.4.

### E.1. Controlling for Task Difficulty: Supervised Pixel Baseline

To verify that pressure remains statistically learnable in the intense regime, we include a supervised pixel-space baseline trained from scratch on cyclone imagery using a ResNet-18 architecture. The model uses 64 base channels, a 128-dimensional hidden layer, and a dropout rate of 0.3, and is optimized with AdamW (learning rate $10^{-4}$, weight decay $10^{-2}$) under a cosine learning-rate schedule with a 100-step warm-up. Training is performed with a batch size of 64 and continued until convergence (within 200 epochs).

This experiment is not intended as a competitive benchmark against frozen foundation models, but rather as a sanity check demonstrating that pressure exhibits a statistically meaningful relationship with the observations even for intense storms. While prediction uncertainty naturally increases for stronger systems, the objective here is simply to verify that the mapping from imagery to pressure is not degenerate under held-out evaluation. For consistency with the main paper, we report mean absolute error (MAE), corresponding to the normalized absolute error used in our alignment probes.

As shown in Fig. 6, the baseline model achieves a normalized mean absolute error of $0.4$ and a normalized median absolute error of $0.3$, normalized by the standard deviation of pressure on the evaluation set, which is consistently lower than the VFM probe results reported in Fig. 2. This indicates that pressure remains statistically predictable from single-frame imagery in the intense regime, even though uncertainty increases, and that increased task difficulty alone does not account for the representational collapse observed in frozen VFMs.

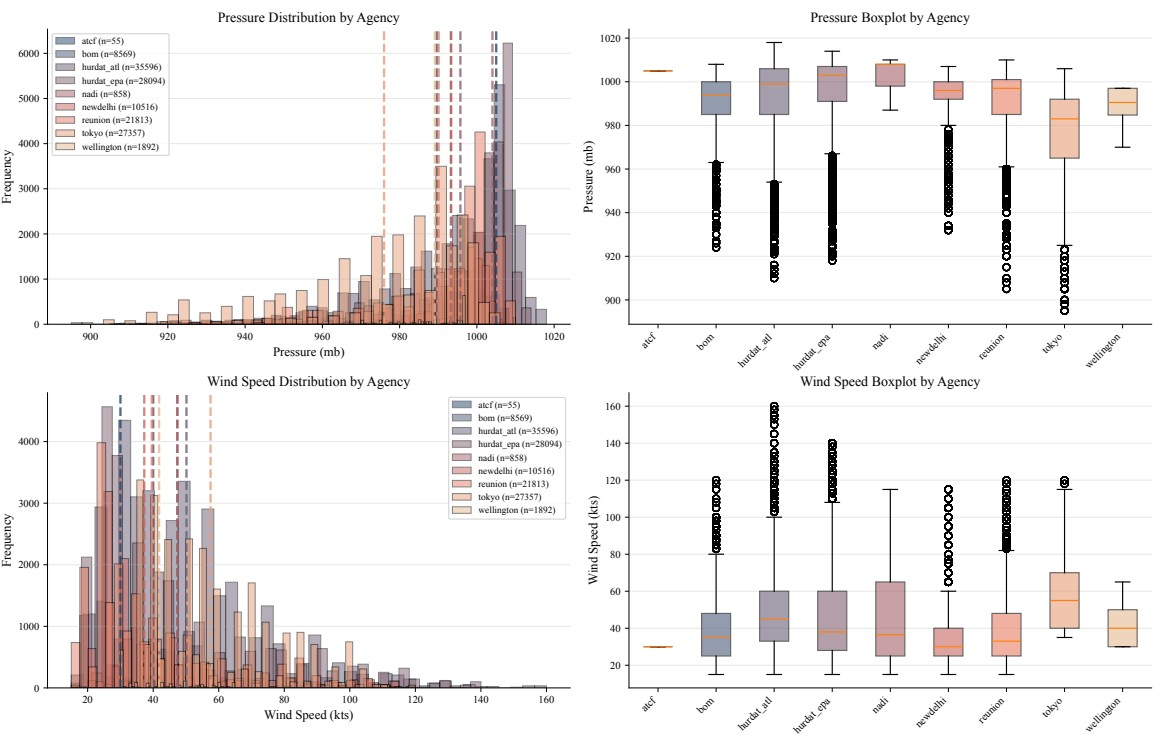

*Figure 5. Dataset inspection*: Per-agency distributions of minimum central pressure $P_c$ (left) and maximum sustained wind speed $V_m$ (right) in the TC-BENCH dataset. While pressure distributions are broadly consistent across agencies, wind speed exhibits noticeable variability due to differences in averaging conventions (e.g., 1-minute vs. 10-minute winds), underscoring the greater robustness of pressure as a cross-agency physical indicator.

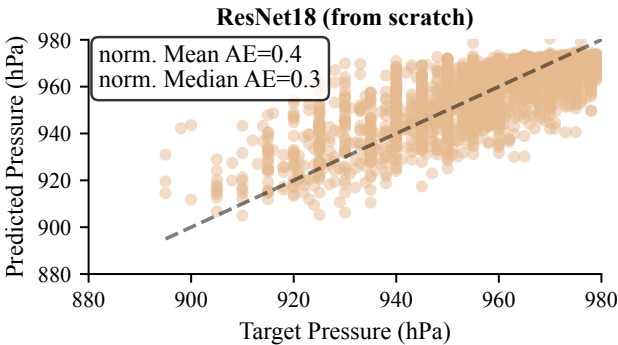

*Figure 6. Supervised pixel baseline.* Predicted versus target pressure for a ResNet-18 trained from scratch and evaluated on held-out intense storms ($P_c < 980$ hPa). In the intense regime, a supervised pixel-space model achieves a normalized mean and median absolute error of 0.4 and 0.3 respectively, indicating that pressure remains statistically predictable from imagery despite increasing uncertainty.

*Table 4.* Nonlinear Static Fidelity probes on DINOv3-base features. Errors are normalized MAE under $Q_{\text{stat}}$, reported as mean ± std over three independent runs. *Increasing probe capacity does not remove the moderate-to-intense degradation.*

| Probe | Moderate | Intense |
|---|---|---|
| 2-layer MLP | $0.382 \pm 0.021$ | $0.473 \pm 0.0327$ |
| Transformer probe | $0.284 \pm 0.020$ | $0.503 \pm 0.022$ |

### E.2. Controlling for Probe Capacity: Nonlinear Probes

While structural isomorphism is defined in terms of linear reparameterization—and is therefore naturally evaluated with linear probes—we additionally test whether the observed regime-dependent degradation can be explained by limited probe capacity. To this end, we replace the linear readout with higher-capacity nonlinear probes, including a two-layer MLP and a lightweight Transformer probe, and re-evaluate Static Fidelity across intensity regimes.

We conduct this analysis on DINOv3-base (Siméoni et al., 2025), which achieves the strongest performance under the linear Static Fidelity probe among the evaluated frozen VFMs (see Fig. 2). As shown in Tab. 4, increasing probe expressivity does not remove the moderate-to-intense degradation: both nonlinear probes exhibit substantially higher normalized error $\xi_{\text{stat}}$ in the intense regime. This indicates that the failure is not merely an artifact of linear readout capacity. Rather, together with the geometric collapse observed in Fig. 4, these results support the interpretation that physically relevant variation becomes poorly resolved in the frozen representation itself.

### E.3. Controlling for Representation Aggregation: Spatial-Token Mean

In addition to the `CLS` token, we evaluate representations obtained by aggregating spatial tokens via a simple spatial mean. Results are shown in Figs. 7 and 8. Consistent with the main findings in § 4, performance degrades consistently in intense regimes, across all model families. This confirms that representation collapse is not specific to the `CLS` token, but is a systematic property affecting both global and spatially distributed features.

### E.4. Controlling for Pretraining Objective: Video and Temporal Models

To test whether the failure is specific to static image pretraining, we evaluate video-based models with temporal predictive or video-language objectives. Representations are extracted from clips using the same trajectory-level splits and regime-balanced evaluation protocol. As shown in Tab. 5, temporal pretraining improves average prediction error but does not remove the moderate-to-intense degradation.

### E.5. Implementation Details.

**Linear Probe.** All linear probing experiments use Ridge Regression implemented in `scikit-learn`. The regularization coefficient $\alpha$ is selected via 5-fold cross-validation over a logarithmically spaced grid from $10^{-3}$ to $10^6$. For the Static

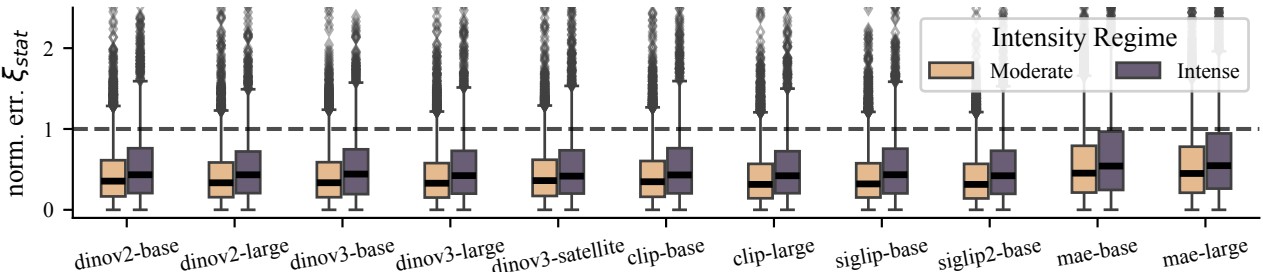

*Figure 7. Static Fidelity error* using the aggregated **spatial mean** of all spatial tokens. As in Fig. 2, performance degrades consistently in intense regimes.

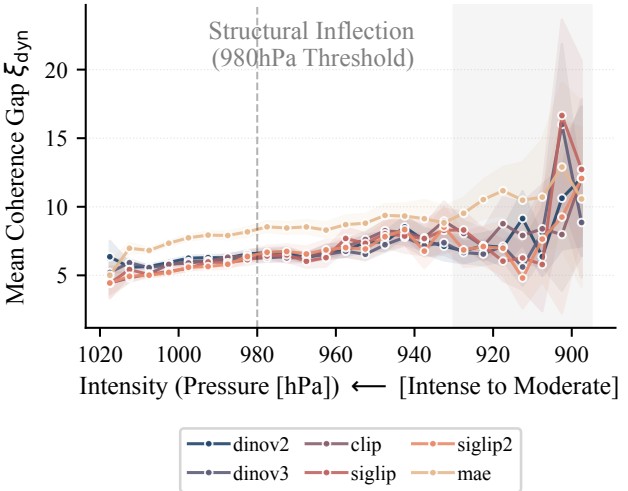

*Figure 8. Dynamic Coherence gap* using the aggregated **spatial mean** of spatial tokens. The coherence error increases with storm intensity, mirroring the behavior observed with `CLS`-based representations.

*Table 5.* Static Fidelity ($Q_{\text{stat}}$) for video-based models. Errors are normalized MAE, reported as mean $\pm$ std over 10 runs. Temporal pretraining improves average prediction but preserves the moderate-to-intense degradation.

| Model | Moderate | Intense |
|---|---|---|
| VideoMAE | $0.333 \pm 0.010$ | $0.434 \pm 0.016$ |
| V-JEPA2 | $0.235 \pm 0.008$ | $0.334 \pm 0.015$ |
| X-CLIP | $0.251 \pm 0.008$ | $0.353 \pm 0.006$ |

Fidelity and Dynamic Coherence probes, we construct a regime-balanced dataset comprising 53,200 samples, while the Manifold Consistency probe uses 47,207 samples. In all cases, the number of samples substantially exceeds the feature dimensionality, ensuring a well-conditioned linear regression problem and avoiding rank-deficient solutions. The $L_2$ regularization enforces stable estimation and enables fair comparisons across representations by controlling overfitting, particularly in regimes with limited effective sample diversity.

### E.6. Feature Visualization

To qualitatively interpret the dominant modes of variation captured by the representation, we visualize the extremes of the first principal component (PC1) of frozen DINOv3 features to probe what dominant variation the representation captures.

In the full dataset, PC1 largely separates visually distinct regimes, contrasting diffuse cloud structures associated with weaker systems against organized spiral patterns characteristic of mature cyclones (Figs. 9b and 9c). However, when restricting to intense storms ($P_c < 950$), the same component no longer corresponds to a clear progression in physical intensity (Figs. 9c and 9d). Images at both extremes of PC1 exhibit highly similar eye and eyewall structures, despite substantial differences in pressure and wind speed. This suggests that, within the intense regime, the dominant axes of variation in the frozen representation are driven primarily by residual visual factors (e.g., orientation, texture, or cloud asymmetries) rather than physically meaningful intensity-related structure. Consistent with our quantitative analysis, this qualitative collapse indicates that physically relevant variation is compressed into a low-dimensional subspace that is misaligned with pressure.

### E.7. Manifold Consistency Probe: Detailed Explanation

We provide additional implementation details for the manifold consistency probe $Q_{\text{con}}$ introduced in § 4.

**Constraint derivation.** Following Chavas & Knaff (2022), both the inner-core structure of a tropical cyclone (e.g., radius of maximum wind) and its outer size tend to increase with latitude. We therefore assume the characteristic storm radius scales approximately as

$$r \sim B\phi,$$

where $\phi$ denotes latitude and $B > 0$ is a constant. Substituting this scaling into the gradient wind balance equation Equation (C.1), and approximating the pressure gradient as $P/r$, yields

$$V^2 + fVr \sim P \iff V^2 + 2\Omega \sin(\phi)\, B\phi\, V \sim P. \tag{E.1}$$

Because the Coriolis term increases with latitude, a lower central pressure is required to sustain the same wind speed at higher latitudes. We therefore define the constraint violation $\psi_{\text{con}}$ as the discrepancy between predicted and observed regional wind differences $\Delta V_m := V_m^{\text{low-lat}} - V_m^{\text{high-lat}}$, which serves as a proxy for gradient wind balance consistency.

**Qualitative results.** To visualize manifold misalignment, we stratify the test set into latitude bands and estimate the conditional relationship $\hat{V}(P)$ via local regression for both ground-truth measurements and model predictions. As shown in Fig. 10, the regional wind separation $\Delta V_m$ increasingly diverges from the physical expectation as forcing intensifies, particularly in the intense regime ($P_c < 980\,\text{hPa}$). This behavior aligns with the Static Fidelity and Dynamic Coherence results, reinforcing that representation collapse emerges precisely where visual cues saturate and cease to reflect the underlying physical state.

## F. Extended Related Work

**Weather Foundation Models.** Recent advances in data-driven weather forecasting have produced high-capacity foundation models (Lam et al., 2023; Price et al., 2025; Alet et al., 2025; Bi et al., 2023; Pathak et al., 2022; Bonev et al., 2025) have increasingly rivaled traditional numerical weather prediction. However, our framework differs significantly as these models typically ingest gridded physical reanalysis fields (e.g., ERA5) to emulate atmospheric evolution, whereas we focus on extracting latent physical states directly from unstructured satellite infrared imagery. Furthermore, while probabilistic models like GenCast (Price et al., 2025) and FourCastNet3 (Bonev et al., 2025) prioritize autoregressive trajectory forecasting and *center tracking*, our work targets Scientific Alignment, ensuring that the model's learned representation is a linear reparameterization of physical invariants like the pressure-gradient relationship, rather than just a predictive emulator.

**Explainable Machine Learning.** Recent efforts in scientific machine learning (SciML) have largely focused on the interpretability and explainability of data-driven models (Roscher et al., 2020). These frameworks typically provide taxonomies

for integrating domain knowledge—either as inductive biases in architecture or as post-hoc explanations of model decisions (Karniadakis et al., 2021; Willard et al., 2022). However, "explainability" in this context can be subjective and often lacks a formal guarantee of physical consistency across out-of-distribution regimes (Geirhos et al., 2020; Rudin, 2019). Our work moves beyond these taxonomies by establishing formal *epistemic criteria* for scientific usability. Instead of treating physics as an optional input, we introduce necessary conditions for *Scientific Alignment* (§ 2) as a set of mathematical prerequisites (Tab. 1), provides a principled way to falsify representational hypotheses about physical and causal interpretability.

**Representation Alignment.** The field of representation alignment primarily focuses on the structural and functional similarities between the internal representations of artificial and biological systems (Muttenthaler et al., 2022; 2025). More recently, this concept has expanded to facilitate latent representation communication (Moschella et al., 2023; Huh et al., 2024; Fumero et al., 2024) and model interoperability through merging and stitching (Ainsworth et al., 2023; Csordás et al., 2021). However, "alignment" in these contexts is typically defined at a purely computational level—measuring the relative distance between different neural encodings—rather than the alignment between an encoding and the underlying domain physics. While existing methods ensure that two models "speak the same language,", they do not guarantee that this language is grounded in scientific reality. In a similar spirit to LLM alignment (i.e., ensuring model outputs adhere to human intent (Ouyang et al., 2022; Gabriel, 2020)), scientific alignment ensures that representations are constrained by physical laws. As we formalize in § 2, this requires moving beyond model-to-model similarity toward a rigorous evaluation of whether a representation supports structural alignment probes $Q$ and surrogate reasoning. This transition from "how models talk to each other" to "how models talk to the world" provides a necessary prerequisite for the reliable employment of data-driven approaches in the natural sciences.

**Physical Common Sense.** Our work relates to recent efforts on evaluating physical reasoning and common-sense understanding in vision and video models. A growing body of benchmarks (Chow et al., 2025; Bear et al., 2021; Zhang et al., 2026; Bansal et al., 2024; Wang et al., 2024; Meng et al., 2024) assesses whether model predictions or generations conform to intuitive physical rules, such as object permanence, collision dynamics, and temporal consistency. These evaluations have provided important insights into perceptual robustness and output-level generalization. In contrast, we focus on a complementary question: whether the *internal representations* learned by vision models preserve the structural organization of the underlying physical system. While strong performance under perceptual or visually defined OOD shifts is often interpreted as evidence of physical understanding, such results do not necessarily constrain the geometry of the latent space itself. Our work therefore emphasizes probing representation structure directly, rather than inferring physical grounding solely from observable outputs.

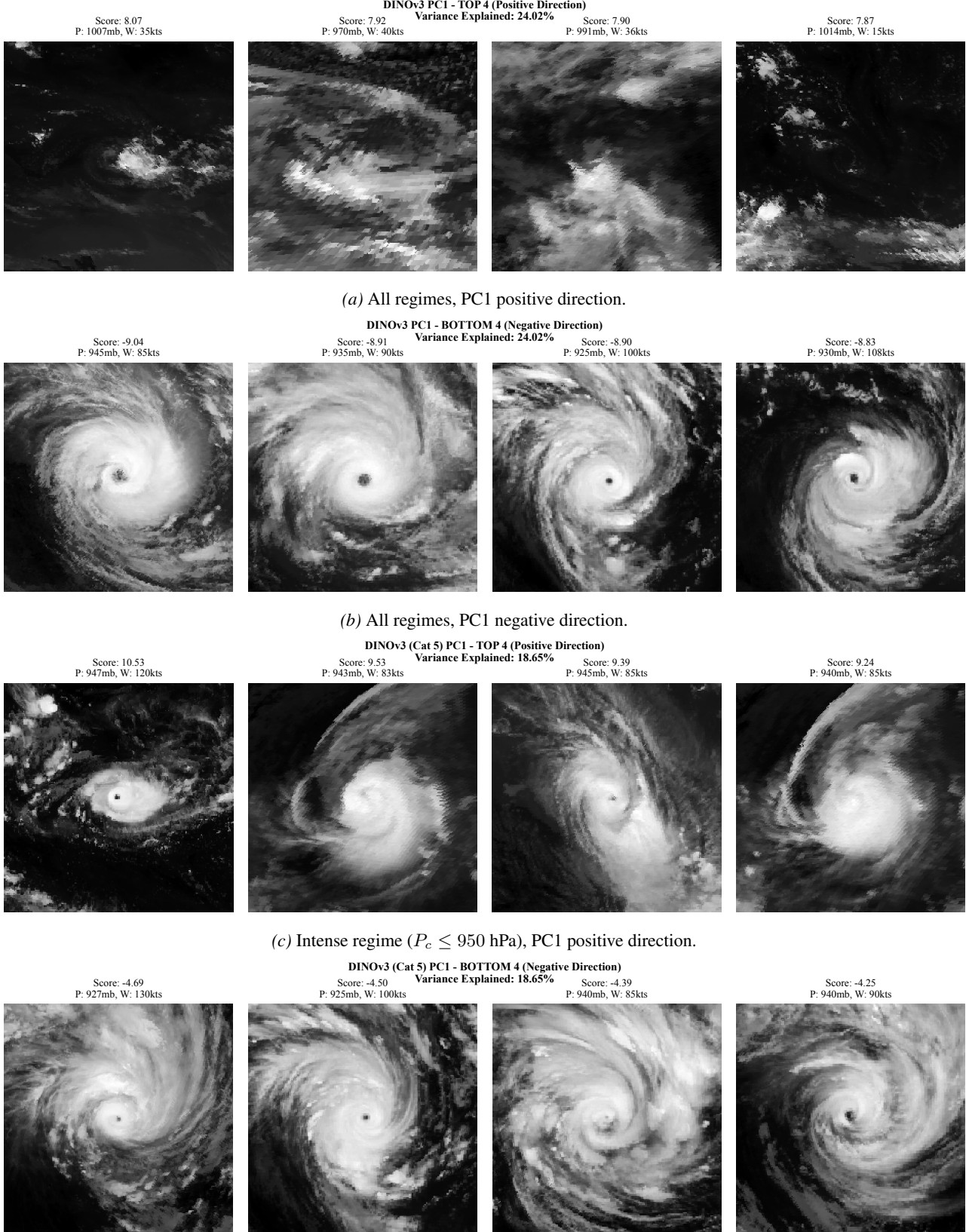

*(a)* All regimes, PC1 positive direction.

*(b)* All regimes, PC1 negative direction.

*(c)* Intense regime ($P_c \leq 950$ hPa), PC1 positive direction.

*(d)* Intense regime ($P_c \leq 950$ hPa), PC1 negative direction.

*Figure 9. PCA exemplar visualization* of the first principal component (PC1) of DINOv3 latent representation.

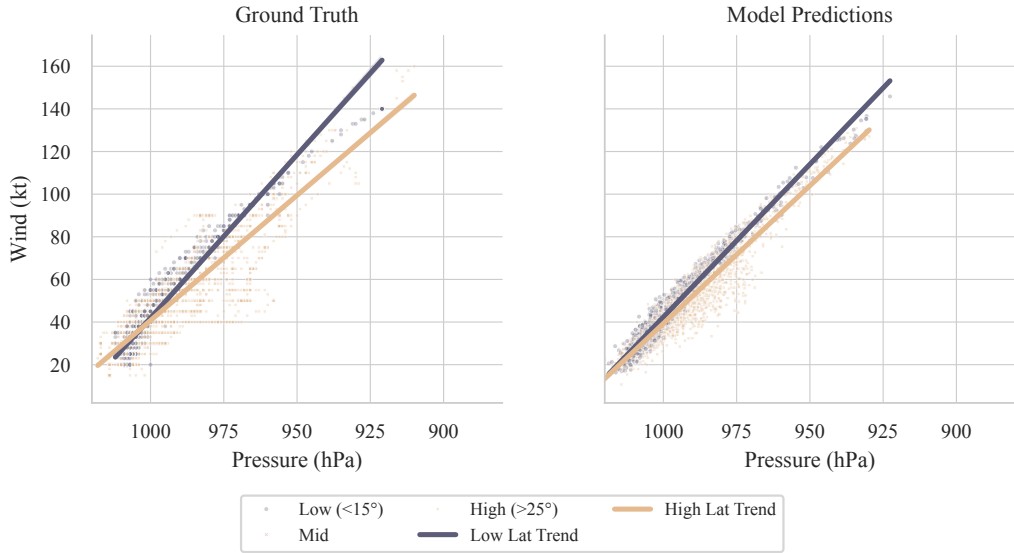

*Figure 10. Manifold inconsistency.* Wind–pressure relationships stratified by latitude. **(A) Ground truth:** Physical constraints imply that low-latitude storms ($< 15°$) require higher winds to sustain the same pressure deficit than high-latitude storms ($> 25°$), producing a clear separation ($\Delta V \approx 15$ kt) in the intense regime. **(B) Model prediction:** The model partially recovers the directional trend but underestimates the magnitude of $\Delta V_m$, indicating a violation of manifold consistency.

