# OpenReview forum: "The Perception–Physics Paradox: Probing Scientific Alignment with TC-Bench"
_ICML.cc/2026/Conference — ICML 2026 regular_

### Official Review · Reviewer_cQqk · 2026-03-09

**Soundness:** 3
**Presentation:** 3
**Significance:** 3
**Originality:** 4
**Overall Recommendation:** 5
**Confidence:** 3

**Summary:**

This work introduces TC-Atlas, a versioned global tropical cyclone dataset with a reproducible construction pipeline, designed to probe whether vision foundation models (VFMs) encode physically meaningful representations of tropical cyclones. The central thesis is the Perception–Physics Paradox: VFMs can generalize perceptually, even performing well on out-of-distribution benchmarks, yet collapse the physical degrees of freedom required for scientific reasoning, particularly in intense storm regimes.

The authors formalize the concept of scientific alignment and operationalize it through structural isomorphism, which requires that a learned latent space identify the physical system up to a linear reparametrization. This framework yields a hierarchy of three probes: (1) Static Fidelity, which tests whether physical state variables (minimum central pressure, P_c) can be linearly recovered from frozen representations; (2) Dynamic Coherence, which tests whether temporal evolution in latent space aligns with physical temporal evolution; and (3) Manifold Consistency, which tests whether independently recovered variables jointly respect known physical couplings.

In TC-ATLAS, the observation space consists of two-dimensional infrared satellite images and the physical state space is defined by minimum central pressure (P_c) and maximum sustained wind speed (V_m). Evaluating a diverse suite of frozen VFMs (DINOv2, DINOv3, CLIP, SigLIP, SigLIP2, MAE), the authors find a consistent pattern across all model families: performance is adequate in the moderate regime (P_c ≥ 980 hPa) but degrades sharply in the intense regime (P_c < 980 hPa). The failure mode analysis reveals that below this threshold, physically distinct cyclones are mapped to a narrow region of latent space, with effective dimensionality dropping and feature spread collapsing. This representational collapse explains why predictions become increasingly inaccurate as the cyclone intensify. Importantly, a supervised pixel-space baseline achieves substantially lower error in the same regime, confirming that pressure-related signal remains present in the observations and that the failure reflects a systematic misalignment in VFM representations rather than inherent task difficulty.

**Compliance With Llm Reviewing Policy:**

Affirmed.

**Final Justification:**

Concerns fully resolved, this is a good work.

**Key Questions For Authors:**

1) Would video-based models improve structural alignment by capturing cyclone evolution dynamics?

2) You mention V-JEPA in related work (Section 5). Did you evaluate Joint-Embedding Predictive Architecture models? If so, what were the results? If not, do you expect JEPA's prediction-based objective to improve physical alignment?

3) Could the scientific alignment probes be integrated into training objectives rather than only post-hoc evaluation?

**Limitations:**

yes

**Strengths And Weaknesses:**

Strengths:

- Compelling problem formulation: The Perception–Physics Paradox is well-articulated, demonstrating that models can achieve strong perceptual OOD generalization while failing to preserve physically meaningful structure.
- Rigorous theoretical framework: The Scientific Alignment framework is formalized through structural isomorphism and yields testable necessary conditions via the probing hierarchy.
- Valuable dataset contribution: TC-ATLAS provides a versioned, global tropical cyclone dataset with a reproducible construction pipeline.
- Excellent physics background: Appendix C offers a thorough introduction to tropical cyclone physics, grounding the work in domain knowledge. I really enjoyed reading it!

Weaknesses:

- Unclear model comparisons: While MAE appears competitive in some results (Figures 2-3), the paper lacks detailed analysis of why specific architectures perform differently. What architectural properties contribute to better/worse physical alignment?
- Limited discussion of failure modes: While Section 4.2 identifies representational collapse, deeper investigation and explanations into what visual features cause this collapse would be valuable.
- It would be better for the reader unfamiliar with the cyclone literature to see some additional samples from the TC-ATLAS dataset as figure in the appendix.

---

> ### Author Rebuttal · Authors · 2026-03-29
>
> We sincerely thank Reviewer `cQqk` for the thorough and constructive feedback, and for the positive assessment. We address each point below.
>
>
> &nbsp;
>
> -------
>
> ### **W1: Architecture Choice and Scientific Alignment**
>
> We thank the reviewer for this question.
>
> **Clarification.**
>
> * MAE does not escape the observed failure mode. While some models may appear competitive on specific metrics, **all evaluated architectures exhibit systematic degradation in the intense regime (Pc \< 980 hPa) under our probing hierarchy.**
> * We additionally evaluated video-based models (`VideoMAE, V-JEPA2, X-CLIP`); **the same regime-dependent collapse persists** (see `Table 1`, response to `Reviewer 3vx4, W3`). This cross-architecture consistency reinforces that the failure is structural, not model-specific.
> * The central message is therefore not about relative ranking, but that **strong perceptual representations across architectures do not necessarily preserve physically meaningful structure.**
>
> **Revision.**
> We agree that understanding architectural biases that may promote scientific alignment is an important future direction, and will clarify this discussion in the revision.
>
>
> &nbsp;
>
> -------
>
> ### **W2: Failure Mode Analysis**
>
> We thank the reviewer for raising this important point.
>
> **Clarification**
>
> * We provide additional insight into the failure mode in the `App. E.3` and `Fig. 9`. Our analysis suggests that VFMs rely heavily on prominent visual cues such as eye-wall and circular cloud structures (see `Fig. 1`). When considering all regimes, the leading principal component (PC1) clearly separates samples with and without a visible eye, as shown by the top positive/negative directions (`Fig. 9a,b`).
> * However, in the intense regime, PC1 continues to focus on variations in eye-wall structure while its explained variance decreases (`Fig. 9c,d`).
> * This indicates that the **representation becomes dominated by a limited set of visually salient features, while failing to capture finer-grained physical variation within the regime.**
>
> **Revision:**
> We agree that further investigation of feature-level mechanisms is valuable and will clarify this analysis in the revision.
>
>
> &nbsp;
>
> -------
>
> ### **W3: Additional Visual Examples of Cyclones**
>
> Thank you for this suggestion. We have provided some cyclone examples based on PCA analysis in `Fig 9`. We are happy to include more diverse examples to further familiarize readers with the TC-Atlas dataset.
>
>
>
> &nbsp;
>
> -------
>
> ### **Q1 & Q2: Evaluating Video Models including V-JEPA**
>
> We have provided additional evaluation results on V-JEPA2 during the rebuttal (see `Table 1`). Please kindly refer to answer `W3` of Reviewer `3vx4` for further explanation.
>
> &nbsp;
>
> -------
>
> ###  **Q3: Scientific Alignment as Training Objectives**
>
> Thanks for this interesting idea\! It could be beneficial to include temporal consistency or constrained learning as either regularization or architectural design into the model, inspired by our dynamical coherence and manifold consistency.  We will add a discussion paragraph regarding this.

---

> > ### Author Rebuttal · Reviewer_cQqk · 2026-04-02
> >
> > Thank you for the rebuttal, my concerns are fully resolved. I am keeping the positive score.

---

### Official Review · Reviewer_YPDw · 2026-03-12

**Soundness:** 3
**Presentation:** 3
**Significance:** 3
**Originality:** 3
**Overall Recommendation:** 5
**Confidence:** 2

**Summary:**

This paper analyzes the Perception-Physics Paradox: when vision foundation models (VFMs) look correct but not reasoning correctly. To this end, the propose scientific alignment, a method to correctly evaluate VFMs, and propose a probing framework dervied from the necessary conditions of scientific alignment. Additionally, they introduce TC-Atlas, a global tropical cyclone dataset which includes intense regimes and lifecycle transitions to enable physicall targeted representational stress tests.

**Compliance With Llm Reviewing Policy:**

Affirmed.

**Final Justification:**

The authors have addressed my concerns in the rebuttal, so I keep my original score.

**Key Questions For Authors:**

1. Could the authors further explain the setting in Fig. 4a (how the experiment was done)? Is the figure indicating that with intense MCP, the latent PC1 will be high? If so, what does it mean?
2. How is "effective dimensionality" measured in Fig. 4b, and how is "feature spread" measured in Fig. 4c?

**Limitations:**

yes

**Strengths And Weaknesses:**

Strengths:
- The paper provides an interesting aspect for current VFMs that their performance comes from visual correlations rather than structural invariants, making OOD accuracy a poor proxy.
- The claims of the paper are well-supported with theoretical analysis and experimental results.
- The paper is clearly written, with good flow which makes it easy for readers to understand and follow.
- In overall, the paper has significant contributions, including the Perception-Physics paradox, scientific alignment as new evaluation framework, and the TC-Atlas dataset.

Weaknesses:
- The analysis is only carried out on the TC-Atlas dataset. It would be better if we provide the results on other datasets to see if the claims hold.
- The paper did not provide/recommend any sort of naive solutions to the modeling problem, it just provide an analysis of the problem of current models and a new evaluation framework.

---

> ### Author Rebuttal · Authors · 2026-03-29
>
> Thank you for the careful review and positive feedback. We address each point below.
>
> &nbsp;
>
> ------
>
> ### **W1: Generalization beyond TC-Atlas**
>
> **Clarification:**
>
> * The Perception–Physics Paradox is not specific to tropical cyclones — it arises whenever **visual observations become insensitive to physical variation in extreme regimes**, while the physical signal remains present in the observations. This is a verifiable, domain-agnostic condition confirmed here by our supervised baseline.
> * **This phenomenon is well-documented beyond cyclones**. *In wildfire remote sensing, brightness temperature observations visually saturate when flames entirely fill a sensor pixel, rendering the observation insensitive to further increases in fire intensity (`Wooster et al., 2005, Sec. 3.2`)* — precisely the saturation mechanism we formalize in TC-Atlas.
> * **Extreme regimes** are systematically underrepresented in existing benchmarks, **yet are where physical reasoning matters most:** cyclones, wildfires, and floods cause disproportionate human and economic impact but are inherently rare and difficult to instrument. Rigorous study of the paradox therefore requires a dataset with **explicit extreme-regime coverage** — the gap TC-Atlas is designed to fill.
>
> **Revision:** We will characterize the structural conditions under which the paradox manifests and include wildfire remote sensing as a concrete additional domain.
>
> *\[1\] Wooster, Martin J., et al. "Retrieval of biomass combustion rates and totals from fire radiative power observations: FRP derivation and calibration relationships between biomass consumption and fire radiative energy release." Journal of Geophysical Research: Atmospheres 110.D24 (2005).*
>
> &nbsp;
>
> ------
>
> ### **W2: On the Solution of Scientific Alignment**
>
> We agree that developing training objectives for improving scientific alignment is a **key next step.**
>
> **Clarification.**
>
> * **The primary goal of this work is diagnostic rather than prescriptive.** Specifically, we aim to challenge the common assumption that strong perceptual generalization implies physically meaningful representations, which we formalize as the *Perception–Physics Paradox*.
> * We believe this diagnostic step is necessary: **without a clear and testable notion of alignment, it is difficult to design or evaluate improved training objectives.** In this sense, our framework provides concrete, testable criteria that can guide and assess future alignment-aware methods.
>
> **Revision:**  We will expand the discussion to highlight concrete directions: *temporal supervision* via physically consistent objectives; *constraint-aware* representation learning; and *multi-view or multi-modal supervision* to better identify latent physical state. We will also reinforce the diagnostic framing in the introduction.
>
> &nbsp;
>
> ------
>
> ### **Q1: Explanation of Figure 4a**
>
> Thank you for this very important question. We really appreciate your careful review\!
>
> **Setup.** For Fig. 4a, we perform a 2-component PCA on the representations and plot PC1 against pressure.
>
> **Observation.** The key observation is that the **variance of PC1 shrinks** as intensity increases.
>
> **Implication.** This **indicates** that in the intense regime, *physically distinct states are mapped to a narrower region along the dominant latent direction,* reflecting a collapse of representation along a physically relevant axis.
>
> **Revision.** We will add a clarifying caption and expanded explanation of this figure to the manuscript.
>
>
> &nbsp;
>
> ------
>
> ### **Q2: Effective Dimensionality and Feature Spread**
>
> **Effective dimensionality** is the participation ratio of the covariance spectrum:
> \\[ d_{\text{eff}} = \frac{(\sum_i \lambda_i)^2}{\sum_i \lambda_i^2}​. \\]
> where \$\lambda\_i$​ are eigenvalues of the within-bin feature covariance. This quantifies how many orthogonal directions contribute significantly to variance.
> * **High d\_eff​:** variance is distributed across many directions (rich, expressive representation).
> * **Low d\_eff​:** variance concentrates in a few directions (**collapse**).
>
> **Feature spread**  is the mean pairwise Euclidean distance between centered features within each pressure bin.  This measures how dispersed representations are locally in feature space.
> * **High spread:** samples remain well-separated (diverse encoding).
> * **Low spread:** distinct physical states become **indistinguishable**.
>
> Together, **both metrics diagnose representation collapse in the intense regime**.
>
> **Revision:** We will add formal definitions and interpretations of both metrics to the manuscript.

---

> > ### Author Rebuttal · Reviewer_YPDw · 2026-04-02
> >
> > I appreciate the authors' detailed responses.
> >
> > The majority of my concerns have been resolved. I decide to keep my positive score.

---

### Official Review · Reviewer_3vx4 · 2026-03-13

**Soundness:** 2
**Presentation:** 3
**Significance:** 2
**Originality:** 2
**Overall Recommendation:** 4
**Confidence:** 2

**Summary:**

This paper studies the gap between perceptual generalization and physical reasoning in vision foundation models (VFMs). The authors show that although these models often achieve strong predictive and OOD performance, they may fail to preserve physically meaningful structure in their latent representations, which is referred to as the Perception–Physics Paradox.
To investigate this, the authors introduce the concept of scientific alignment and present TC-ATLAS, a global tropical cyclone dataset with a reproducible construction pipeline. Using this dataset, they evaluate several VFMs (e.g., DINO, CLIP, SigLIP, MAE) through probing analyses. The results suggest that strong performance on standard benchmarks does not necessarily translate to physically meaningful representations.

**Compliance With Llm Reviewing Policy:**

Affirmed.

**Final Justification:**

The reply to my rebuttal acknowledgement clarified the intent of the paper and addressed my main concerns. Based on this I have increased my score.

**Key Questions For Authors:**

1. Related to Weakness 3, how would models trained specifically for physical prediction perform under the proposed evaluation framework?
2. Since the evaluation framework relies on linear probes, it would be helpful to understand how sensitive the results are to the choice of probe. Could the authors clarify whether the conclusions depend on the assumption that physically meaningful variables should be linearly identifiable from the learned representations?

**Limitations:**

Yes

**Strengths And Weaknesses:**

Strengths
1. The paper is well written and easy to follow. Structural isomorphism and structural alignment probes are clearly defined and well motivated.
2. The paper raises an important question for scientific ML and foundation models: whether strong perceptual performance implies representations aligned with the underlying physical system. The distinction between predictive accuracy and faithful representation is an important issue for scientific applications.
3. The paper presents an interesting observation that strong OOD generalization does not necessarily imply scientifically meaningful representations.

Weaknesses
1. The paper mainly identifies the problem but does not propose a concrete solution. In particular, it does not provide a training method or representation learning objective that could improve scientific alignment.
2. The experiments are conducted on a single dataset within one scientific domain. It remains unclear whether the findings generalize to other scientific settings.
3. The models used for evaluation do not include models specifically trained for physical prediction, therefore the observed behavior may partly reflect biases from general image pretraining.
4. The evaluation primarily relies on linear probes. It remains unclear whether the failure of linear probes reflects representational misalignment or simply nonlinear encoding of the physical variables.

---

> ### Author Rebuttal · Authors · 2026-03-29
>
> We sincerely thank the reviewer for the thoughtful and constructive feedback. Below, we clarify several points and outline our revisions to the paper. (Due to character limits, we kindly invite you to find the detailed responses for `W1 and W2` in the rebuttal for `Reviewer YPDw`.)
>
> &nbsp;
>
> ---
>
> ### **W1: On the Solution of Scientific Alignment**
>
> Please see `Reviewer YPDw (W2)` for our full response.
>
> &nbsp;
>
> ---
>
> ### **W2: Generalization beyond TC-Atlas**
>
> Please see `Reviewer YPDw (W1)` for our full response.
>
> &nbsp;
>
> ---
>
> ### **W3: Physical Prediction vs. Image Pretraining Bias**
>
> We thank the reviewer for this important point. We agree that models trained with explicit physical supervision could behave differently.
>
> **Clarification.**
>
> We take a step in this direction by evaluating video-based models with **temporal predictive objectives** (VideoMAE [1], V-JEPA2 [2], X-CLIP [3]), which incorporate stronger supervision than static image models. Representations are extracted from clips $(x_{t−15}, …, x_t)$ using identical trajectory-level splits and regime-balanced evaluation.
>
> **Result.** Temporal pretraining *does improve overall prediction error* — confirming that additional inductive biases, such as temporal supervision, can help. However, **the moderate-to-intense degradation (\~30% increase in normalised error) persists consistently across all models** (`Table 1`).
>
> **Interpretation.**
>
> Under `Defn. 2.3`, scientific alignment requires uniform performance across regimes — average improvement is not sufficient. **The persistent regime gap means these models remain misaligned precisely where physical reasoning matters most.** The Perception–Physics Paradox, therefore, reflects a limitation of perceptual objectives broadly, not a bias of any particular architecture.
>
> **Revision.** We will include these results in Table 1\.
>
> ---
>
> **Table 1**: Normalized MAE (error / σ(P_c)) under Q\_stat, mean ± std over 10 runs.
>
> | **Model** | **Moderate** | **Intense** |
> | ----- | ----: | ----: |
> | `VideoMAE` | 0.333 ± 0.010 | 0.434 ± 0.016 |
> | `V-JEPA2` | 0.235 ± 0.008 | 0.334 ± 0.015 |
> | `X-CLIP` | 0.251 ± 0.008 | 0.353 ± 0.006 |
>
> *[1] Tong, Zhan, et al. "Videomae: Masked autoencoders are data-efficient learners for self-supervised video pre-training." Advances in neural information processing systems 35 (2022): 10078-10093.*
>
> *[2] Assran, Mido, et al. "V-jepa 2: Self-supervised video models enable understanding, prediction and planning." arXiv preprint arXiv:2506.09985 (2025).*
>
> *[3] Ni, Bolin, et al. "Expanding language-image pretrained models for general video recognition." European conference on computer vision. Cham: Springer Nature Switzerland, 2022.*
>
>
> &nbsp;
>
> ---
>
> ### **W4: Nonlinear-Probe**
>
> We thank the reviewer for this helpful suggestion.
>
>
> **Clarification.**
>
> * The goal of our probes is to assess whether physically meaningful structure is *directly accessible* in the representation. Linear probes are therefore a natural diagnostic: they correspond to a change of basis, and strong performance indicates that physical variables align with latent axes.
> * **If high-capacity nonlinear probes are required, this suggests that physical structure is encoded implicitly rather than structurally,** indicating misalignment under our definition (`Defn. 2.3`).
>
> **Empirical evidence.** We further verify empirically that our conclusion does not depend on linearity, evidenced by the following:
> * A 2-layer MLP (`App. E.1, Fig. 6b`) shows similar regime degradation.
> * A Transformer probe (lightweight, 2-layer encoder, hidden dim 128, 4 attention heads) confirms consistent trends (`Table 2`).
> * `Section 4.2` further shows a *clear drop in effective dimensionality* in the intense regime — a geometric property of the representation independent of probe choice.
>
> **Table 2**: Normalized MAE (error / σ(P\_c)) under Q\_stat, stratified by intensity regime (mean ± std over 3 independent runs).
>
> |  | **Moderate** | **Intense** |
> | ----- | ----: | ----: |
> | `Transformer Probe` | 0.284 ± 0.020 | 0.503 ± 0.022 |
>
> **Revision.**
>
> We will include the Transformer probe results and clarify this point in the manuscript.
>
>
> &nbsp;
>
> ---
>
> ### **Q1: Evaluating Physical-Prediction Models**
>
> We thank the reviewer for this question. Our video model evaluation (see `W3, Table 1`) takes a step in this direction by evaluating models with *temporal predictive objectives*, and shows the paradox persists even under stronger supervision.
>
> &nbsp;
>
> ---
>
> ### **Q2: Dependence on the Linearity Assumption**
>
> We kindly clarify that *our conclusions do not rely on linearity* — both nonlinear probes and the geometric failure mode in `Section 4.2` confirm that the **failure is driven by representational collapse rather than probe capacity**. Please see `W4` for the full empirical evidence and theoretical justification.

---

> > ### Author Rebuttal · Reviewer_3vx4 · 2026-04-02
> >
> > Thank you for the detailed response and for pointing me to the additional clarifications for W1 and W2. The rebuttal improved the clarity of the paper.
> > I appreciate the added experiments addressing W3 and W4, particularly the inclusion of temporal predictive objectives and nonlinear probes, which strengthen the empirical analysis.
> > Regarding W1 and W2, I understand the authors’ positioning of the work as primarily diagnostic and the argument that it may extend beyond TC-Atlas. However, these points rely on conceptual arguments, and their generalizability and practical impact remain unclear. While the diagnostic step is valuable, the work lacks a concrete methodological contribution.
> > I also feel that the key points for W1 and W2 could have been briefly summarized in this response, rather than requiring the reader to refer to another reviewer’s rebuttal. Including a brief summary here would make the rebuttal more self-contained.
> > Since my main reservations about generalization and the lack of actionable solutions remain, I will keep my score.

---

> > > ### Author Response · Authors · 2026-04-04
> > >
> > > We sincerely apologize for the cross-referencing in our rebuttal. While this approach is consistent with the ICML rebuttal guidelines under strict character limits, we fully understand it may introduce additional effort for the reader, and we thank the reviewer for carefully reviewing our responses.
> > >
> > > Please allow us to provide two clarifications regarding W1 and W2:
> > >
> > > + **On methodological contribution.** We agree that methodological advances toward scientific alignment are important. **Meaningful method development, however, requires a precise problem formulation**. Our work identifies and operationalizes a key failure mode: strong perceptual OOD performance does not guarantee physically meaningful latent structure. We make this concrete through **explicit testable criteria—static fidelity, dynamic coherence, and manifold consistency—providing a principled basis for developing and evaluating future methods.**
> > >
> > > + **On generalization**. Our objective is to establish this failure mode in a realistic scientific setting, rather than to claim universality across all domains. **Demonstrating this failure mode in a real-world domain of high scientific and societal importance—tropical cyclones—highlights a critical gap between perceptual generalization and physically meaningful reasoning in current models**. More broadly, the issue arises when visual observations become insensitive to physical variation in extreme regimes. This condition also appears in other scientific sensing settings, such as wildfire intensity saturation, indicating that **the mismatch between perceptual and physical generalization is not specific to TC-Atlas**.
> > >
> > > We hope this clarifies our intent, and we would be happy to further clarify any remaining concerns.

---

### Decision · Program_Chairs · 2026-04-30

**Decision:**

Accept (regular)

**Comment:**

The paper investigates the "Perception-Physics Paradox" in vision foundation models: models that perform well may fail to preserve physically meaningful structure in their latent spaces. This paper formalizes the problem and produces a scientific dataset to study it (TC-ATLAS, a tropical cyclone dataset).

All three reviewers find the problem formulation compelling and the paper clearly written. They also value the TC-ATLAS dataset and the physics background. The main weaknesses raised are: (i) the study is restricted to a single scientific domain (tropical cyclones), so it's unclear whether the paradox holds in broader contexts, and (ii) the contribution is primarily diagnostic (no new method is proposed to improve scientific alignment).

Overall, two reviewers recommend acceptance with high confidence. One reviewer gives a weak accept, primarily due to the lack of a methodological contribution and limited evaluation.